# Noncanonical circRNA biogenesis driven by alpha and gamma herpesviruses

Sarah E Dremel[1,7], Vishal N Koparde [2,3], Jesse H Arbuckle [4], Chad H Hogan [1,5,8], Thomas M Kristie [4], Laurie T Krug [1], Nicholas K Conrad [6] & Joseph M Ziegelbauer [1✉]

## Abstract

**Herpesviruses require the host transcriptional machinery, inducing significant changes in gene expression to prioritize viral transcripts. We examined alpha- and gamma-herpesvirus alterations to a type of alternative splicing, namely circular RNA (circRNA) synthesis. We developed "Circrnas in Host And viRuses anaLysis pIpEline" (CHARLIE) to facilitate viral profiling. This method identified thousands of back-splicing variants, including circRNA common to lytic and latent phases of infection. Ours is the first report of Herpes Simplex Virus-1 circRNAs, including species derived from ICP0 and the latency-associated transcript. We characterized back-splicing *cis*- and *trans*-elements, and found viral circRNAs resistant to spliceosome perturbation and lacking canonical splice donor-acceptors. Subsequent loss-of-function studies of host RNA ligases (RTCB, RLIG1) revealed instances of decreased viral back splicing. Using eCLIP and 4sU-Sequencing, we determined that the KSHV RNA-binding protein, ORF57, enhanced synthesis for a subset of viral and host circRNAs. Our work explores unique splicing mechanisms driven by lytic infection, and identifies a class of transcripts with the potential to function in replication, persistence, or tumorigenesis.**

**Keywords** Circular RNAs; Herpesviruses; Alternative Splicing; RNA-Binding Proteins
**Subject Categories** Microbiology, Virology & Host Pathogen Interaction; RNA Biology

## Introduction

The *Orthoherpesviridae* is an extensive virus family with host-adapted species for a range of organisms including birds, reptiles, and mammals. There are nine herpesvirus species known to infect humans, including the alphaherpesvirus Herpes Simplex Virus-1 (HSV-1, human herpesvirus 1), and the gamma-herpesvirus Kaposi sarcoma herpesvirus (KSHV, human herpesvirus 8). HSV-1 commonly causes recurrent oral and genital lesions. The virus is also responsible for more severe diseases, including herpes keratitis, neonatal mortality and congenital malformations, and viral encephalitis (Bernstein et al, 2013; Tullo, 2003; Tyler, 2018). Individuals are usually infected with HSV-1 during childhood, and by adulthood >40% of the global population tests seropositive (James et al, 2020). KSHV is the etiological agent of several cancers, including Kaposi sarcoma and primary effusion lymphoma (Cesarman et al, 2019). Additional KSHV-associated pathologies include Multicentric Castleman disease (MCD) and KSHV inflammatory cytokine syndrome (KICS). KSHV seroprevalence varies by region and is highest in sub-Saharan Africa (>50%), the Mediterranean (20–30%), and among individuals living with HIV/AIDS in the U.S (~40%) (Cesarman et al, 2019; Labo et al, 2015). A closely related virus, murine gamma-herpesvirus 68 (MHV68), has high genetic homology to KSHV and serves as a tractable animal model for chronic infection (Dong et al, 2017; Wang et al, 2021). Despite decades of study, a therapeutic agent capable of clearing these viruses is lacking. Of the nine human herpesviruses, only varicella zoster virus has FDA-approved vaccines.

Herpesviruses have linear, double-stranded, DNA genomes and replicate within the host nucleus, requiring the host transcription machinery for synthesis of their >100 viral RNAs (Alwine et al, 1974). Like their host, herpesviruses express multiple types of RNAs, including, messenger RNA (mRNA), long noncoding RNA (lncRNA), microRNA (miRNA), and a recently emerging class of transcripts—circular RNAs (circRNA). CircRNAs are single-stranded RNA species formed by a 5' to 3' covalent linkage. In 2018, three groups published the discovery of virus-derived circRNA (Tagawa et al, 2018; Toptan et al, 2018; Ungerleider et al, 2018). Viral circRNAs have now been experimentally confirmed for herpesviruses, papillomavirus, polyomavirus, hepatitis B virus, respiratory syncytial virus, and coronaviruses (Abere et al, 2020; Cai et al, 2020; Huang et al, 2019; Sekiba et al, 2018; Tagawa et al, 2018; Toptan et al, 2018; Ungerleider et al, 2018; Ungerleider Nathan et al, 2018; Yang et al, 2022, 2024; Yao et al, 2021; Zhao et al, 2019). In addition, bioinformatic evidence

¹HIV and AIDS Malignancy Branch, National Cancer Institute, Bethesda, MD 20892, USA. ²CCR Collaborative Bioinformatics Resource, Center for Cancer Research, National Cancer Institute, National Institutes of Health, Bethesda, MD 20892, USA. ³Advanced Biomedical Computational Sciences, Frederick National Laboratory for Cancer Research, Leidos Biomedical Research, Inc, Frederick, MD 21701, USA. ⁴Laboratory of Viral Diseases, National Institute of Allergy and Infectious Diseases, Bethesda, MD 20892, USA. ⁵Graduate Program in Genetics, Stony Brook University, Stony Brook, NY 11794, USA. ⁶Department of Microbiology, UT Southwestern Medical Center, Dallas, TX 75390, USA. ⁷Present address: Department of Microbiology, Immunology and Cancer Biology, University of Virginia, Charlottesville, VA 22908, USA. ⁸Present address: Institute for Genomic Health, Icahn School of Medicine at Mount Sinai, New York, NY 10029, USA. ✉E-mail: ziegelbauerjm@nih.gov

supports expression of circRNAs from other viral genomes (Cai et al, 2021; Fu et al, 2023). Host circRNAs have been shown to function as miRNA sponges, protein scaffolds or sponges, transcriptional enhancers, and translational templates (Kristensen et al, 2019). Less is known regarding the function of viral circRNAs; some serve as translational templates and are packaged in virions and extracellular vesicles (Abere et al, 2020; Zhao et al, 2019). Gamma-herpesvirus circRNAs have been shown to modulate cell growth and apoptosis, pathways of particular interest for oncogenic viruses (Tagawa et al, 2018; Tagawa et al, 2021a). CircRNAs have also been proposed to be putative biomarkers for cancer, with cancer-specific expression profiles and a global anti-correlation between circRNA steady-state levels and cell proliferation (Chen and Shan, 2021; Patop and Kadener, 2018). In line with this, our lab detected KSHV circRNAs in lymph node biopsies obtained from primary effusion lymphoma patients (Tagawa et al, 2021a). CircRNAs are currently being investigated as a mechanism for mRNA-vaccine delivery as they have low immunogenicity, long half-lives, and the potential for cap-independent translation (Enuka et al, 2016; Pamudurti et al, 2017; Qu et al, 2022).

The canonical mechanism of eukaryotic circRNA synthesis, back splicing, is catalyzed by the spliceosome and promoted by RNA-binding proteins (RBPs) or tandem repeat elements which mediate the interaction of back splice junction (BSJ) flanking sequences (Ashwal-Fluss et al, 2014; Conn et al, 2015; Jeck et al, 2013; Li et al, 2017b; Starke et al, 2015). A spliceosome-independent mechanism of circRNA synthesis was identified in archaea (*H. volcanii*) and metazoa (*D. melanogaster*) (Lu et al, 2015; Salgia et al, 2003). Here, the tRNA splicing machinery creates circularized tRNA introns (tricRNA) via tRNA splicing endonuclease (TSEN)-mediated intron cleavage and RTCB catalyzed 3' to 5' RNA ligation. Additional methods for spliceosome-independent circularization have been published for in vitro circRNA synthesis (Litke and Jaffrey, 2019; Obi and Chen, 2021). These leverage either chemical or enzymatic ligation. In the case of enzymatic ligation, T4 DNA ligase (with a complementary ssDNA splint), T4 RNA ligase, or RTCB can circularize RNA. Spliceosome-dependent back splicing may apply to herpesvirus circRNAs like KSHV circvIRF4 and Epstein Barr Virus (EBV) circRPMS1 which are derived from multi-exon genes with flanking splice donor–acceptor sites (Liu et al, 2019; Toptan et al, 2018). However, the structure of these viral genes is not the norm, but rather the exception. Most herpesvirus genes are short (0.5–1.5 kb), single exon, and overlapping or abutting other genes. In addition, HSV-1 inhibits the spliceosome during lytic infection (Hardy and Sandri-Goldin, 1994; Tang et al, 2016). CircRNA species have also been identified for cytoplasmic-replicating RNA viruses (Cai et al, 2020; Yao et al, 2021), a finding seemingly incompatible with spliceosome-dependent mechanisms. Whether circRNAs derived from RNA or DNA viruses proceed through canonical (spliceosome) or noncanonical mechanisms is unclear.

As alternative splicing products, circRNA share almost complete sequence complementarity with their linear counterpart derived from the same gene. The only unique sequence is their 5' to 3' BSJ. High-throughput sequencing (HTS) paired with chimeric transcript analysis enables global circRNA detection and quantification (Gao et al, 2015; Ma et al, 2021). These techniques have revealed that circRNAs are ubiquitously expressed in an array of organisms and tissues (Maass et al, 2017; Salzman et al, 2013). Herein, we characterize the circRNAome of the human herpesviruses HSV-1 and KSHV, as well as that of MHV68, a mouse model for KSHV. We used cell culture and mouse models to thoroughly profile circRNAs expressed during lytic, latent, and reactivation phases. A custom bioinformatic pipeline, called CHARLIE (Circrnas in Host And viRuses anaLysis pIpEline) was developed to facilitate high-throughput analysis of viral circRNAs. Next, we characterized *cis*- and *trans*-acting factors which promote circRNA biogenesis. We found that >90% of viral circRNA species lacked canonical splice donor–acceptor sites. Hotspots of viral back splicing were subsequently found refractory to inhibition of the spliceosome or lariat debranching enzyme. Thus, we tested the potential contribution of alternative *trans*-acting factors, namely RNA ligases. Global profiling of viral back splicing found a significant decrease in viral circRNAs after RNA ligase depletion (RLIG1 knockdown-HSV-1; RLIG1 or RTCB knockdown-KSHV). These data support RNA ligases as novel *trans*-acting factors in viral circRNA synthesis, with loci-specific dependencies. Finally, using eCLIP and Nascent (4sU) RNA-Seq, we determined that the KSHV RNA-binding protein (ORF57) enhanced circRNA synthesis for a subset of viral and host transcripts. Our work identifies dozens of novel herpesvirus transcripts and identifies contrasting circRNA characteristics and synthesis machinery for those derived from the host versus viral genome.

## Results

### Herpesvirus circRNA repertoire

We used models spanning primary lytic infection, latency, and reactivation to explore circRNAs expressed during these distinct transcriptional programs (Fig. 1A; Appendix Fig. S1). HSV-1 models include fibroblasts (MRC-5), trigeminal ganglia (TG), and TG explants (TG explant) with or without a reactivation enhancer (JQ1). KSHV models include lymphatic or vascular endothelial cells (LEC or VEC) and iSLK-BAC16 with or without reactivation enhancers (doxycycline, Dox; sodium butyrate, NaB). MHV68 models include fibroblasts (3T3), splenic germinal center B cells (GC B-cell), and A20 HE-RIT (HE-RIT) with or without reactivation enhancers (Dox; tetradecanoyl phorbol acetate, TPA). If indicated, total RNA was digested with ribonuclease R (RNase R) to enrich circRNAs prior to performing RNA-sequencing (RNA-Seq). CircRNAs were identified by their unique sequence element, the back splice junction (BSJ), a 5' to 3' covalent linkage that generates the continuous loop. Our custom pipeline, CHARLIE (Circrnas in Host And viRuses anaLysis pIpEline), performs de novo circRNA calling for small genome assemblies (e.g., viruses) and does not require canonical flanking *cis*-elements (e.g., splice donor–acceptor). Additional stringency filters are applied to determine high-confidence BSJ, including consistent calls between RNA-Seq mapping algorithms (STAR, BWA) and chimeric junction tools (CIRCexplorer2 + one more). Using this approach, we identified thousands of high-confidence BSJs with significant overlap between biological replicates (Appendix Fig. S2A). Our findings are summarized in a resource table with the genomic position, BSJ flanking sequence, splice donor–acceptor, and raw read count per condition for all high-confidence viral BSJs (Dataset EV1). Overall mapping statistics and circRNA calls are reported in Appendix Tables S1 and S2.

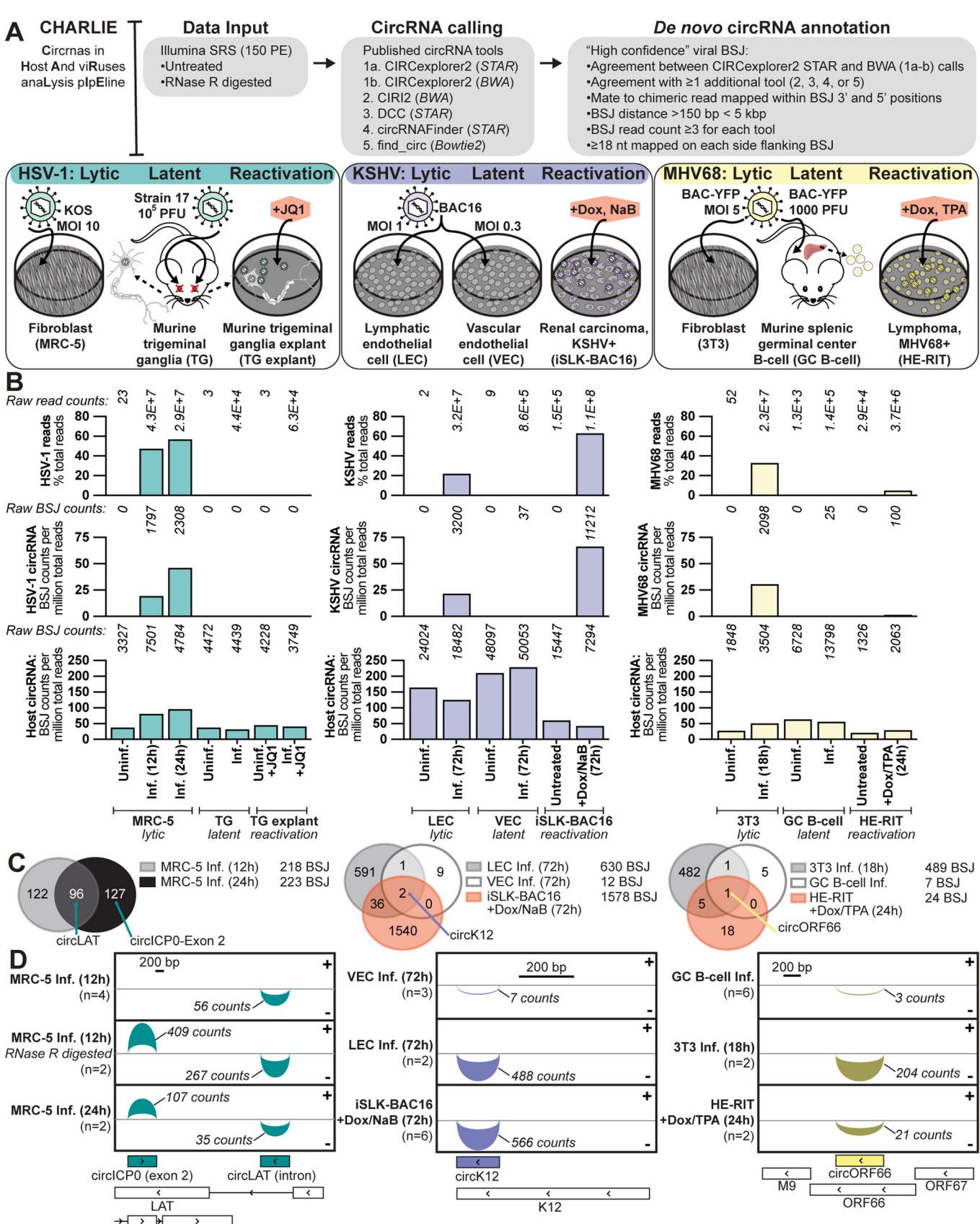

**Figure 1.   Herpesvirus circRNA repertoire.**

(A–D) Infection models include HSV-1: infected fibroblasts (MRC-5), latently infected murine trigeminal ganglia (TG), and latently infected murine TG explanted (TG explant) with or without a reactivation enhancer (JQ1); KSHV: infected human dermal lymphatic endothelial cells (LEC) or human umbilical vascular endothelial cells (VEC), iSLK-BAC16 reactivated with doxycycline (Dox) and sodium butyrate (NaB); MHV68: infected mouse fibroblasts (3T3), latently infected murine germinal center B cells (GC B-cell), and A20 HE-RIT (HE-RIT) reactivated with Dox and tetradecanoyl phorbol acetate (TPA). (A) Infographic for profiling strategy. (B) All reads mapped to the viral genome as a percentage of total reads. The sum of raw counts for each condition are reported above. High-confidence viral or host circRNAs plotted as BSJ counts per million total reads. The sum of raw BSJ counts for each condition are reported above. Column bars are the average of biological replicates. (C) Venn diagrams of overlapping high-confidence viral back splice junction (BSJ) variants. (D) Sashimi plots for select high-confidence circRNA variants, arcs are proportional to raw BSJ counts. Data Information: RNA-sequencing was performed, and high-confidence circRNAs were called using CHARLIE (Circrnas in Host And viRuses anaLysis pIpEline).

Viral circRNA abundance echoed total read composition, with the highest number detected during lytic infection (Appendix Tables S1 and S2; Fig. 1B). Host circRNA expression was less dynamic and largely influenced by cell type; with endothelial cells (LEC, VEC) possessing the most BSJ reads (Fig. 1B). During primary lytic infection, 218, 630, and 489 unique BSJ variants were expressed by HSV-1, KSHV, and MHV68, respectively (Fig. 1C). We compared the repertoire of viral BSJ variants expressed in our models. Notably, KSHV circK12 and MHV68 circORF66 were expressed in lytic, latent, and reactivation models (Fig. 1C,D). To validate our sequencing results, we used divergent primer amplification and RNase R resistance assays. In this assay, linear molecules with 5' and 3' ends are substrates for the endonuclease and degraded, while circularized molecules are resistant to digestion. As expected, RNase R digestion resulted in an enrichment of BSJ reads detected via sequencing (Appendix Table S2). In Appendix Fig. S2B, we evaluated the overlap between untreated and RNase R-treated samples, with 102 out of 218 HSV-1 BSJ (47%) and 80 out of 630 KSHV BSJ (13%) in common (Appendix Fig. S2B). In the same samples, 311 out of 924 (34%) and 1358 out of 2664 (51%) human BSJ were in common between untreated and RNase R-treated samples. All herpesvirus circRNAs measured via qPCR were either resistant or enriched after RNase R digestion (Appendix Fig. S3A). In Appendix Fig. S3B, we visualized the size of divergent primer amplicons relative to the expected product size predicted by CHARLIE. These amplicons were sequenced and had high sequence match (85–100% identity) with the variant identified by RNA-Seq (Appendix Fig. S3B–E). The amplicon products from all divergent primer sets used in this study were sequenced and reported in Dataset EV2.

No HSV-1 BSJ variants were detected in murine models with our standard cutoff of ≥3 BSJ reads (Fig. 1C; Appendix Table S2). As viral gene expression is particularly restricted during latency (Appendix Fig. S1A), we lowered our cutoff to ≥1 BSJ reads and reanalyzed high-confidence circRNAs. With the lower threshold, we identified a dozen BSJ variants derived from the HSV-1 latency-associated transcript (LAT) (Fig. EV1A). Of these, circLAT_KT899744.1: 6137-6814 was expressed in murine trigeminal ganglia and human fibroblasts (Fig. EV1B). circLAT_KT899744.1: 6137-6814 is contained with the stable lariat intron of LAT, with the 3' end of the BSJ matching the 5' end of the intron. We validated circLAT via divergent primer amplification and electrophoresis (Fig. EV1C) or sequencing (Fig. EV1D,E). circLAT was also tested via RNase R resistance assays in a primary lytic infection (MRC-5 12 hpi) and latent infection (murine TG 4 wpi) model (Fig. EV1F). These experiments corroborate the identity of circLAT_KT899744.1: 6137-6814 identified from our RNA-Seq

method. Our profiling identifies novel viral transcripts with distinct modes during infection programs and is the first report of HSV-1 circRNAs.

## Profile of high-incidence lytic circRNAs

We next examined high-confidence circRNAs detected in primary lytic models (Fig. 2A). Viral circRNA traces mimicked that of linear transcripts, with select genes being hotspots (>50 raw BSJ reads) for back splicing (Fig. 2). These included circRNAs colinear with HSV-1 ICP0, LAT, UL12, UL19, UL36, and UL42; KSHV PAN, vIRF4, and K12; and MHV68 ORF66 (Fig. 2B–D; Dataset EV1). During HSV-1 lytic infection, we detected circularization of the middle exon of ICP0 and a truncated version of the LAT lariat intron (Fig. EV2A). There was also a highly abundant circRNA derived from the HSV-1 processivity factor, UL42 (Fig. EV2A). A comparison of circRNA traces from KSHV primary infection (LEC) and lytic reactivation (iSLK-BAC16) revealed overlapping hotspots, e.g., K2, K4, PAN, vIRF4, ORF58-59, K12 (Fig. 2C). This observation contrasts with the ~6% overlap in viral BSJ detected in LEC and iSLK-BAC16 models (Fig. 1C). We would like to note the difference between BSJ position and circRNA sequence content. For instance, a BSJ skewed 1 nt upstream would be counted as a distinct species in Fig. 1C even if the total sequence content shared 99% identity with adjacent species. The pattern of KSHV circRNA detected in LEC and iSLK-BAC16 models (Fig. 2C) suggests that the total sequence content of KSHV lytic circRNAs is similar across different cell types, even if the exact back-splicing junction is variable. Consistent with prior profiling (Tagawa et al, 2018; Toptan et al, 2018; Ungerleider Nathan et al, 2018), major circRNA species expressed by KSHV include a BSJ variant within vIRF4 and a BSJ cluster in PAN (Figs. 2C and EV2B). We also identified an additional high-incidence species, colinear to KSHV K12. Four BSJ variants comprised this cluster, with a single 3' splice acceptor (NC_009333.1: 117534) and a slightly variable 5' donor (NC_009333.1: 117686, 117687, 117688, 117689). Low abundance circK12 species were previously detected in reactivated B-cell models (Toptan et al, 2018); however, the species identified in our adherent cell models is distinct from the previously described circRNA. The major circRNA species expressed by MHV68 was a cluster within ORF66-69, with less abundant circRNAs expressed from ORF17 and ORF75A-C. The cluster within MHV68 ORF66-69 is consistent with findings from Ungerleider et al (2018).

To determine copy levels of select high-confidence circRNAs, we performed digital droplet PCR (ddPCR) using divergent primers which target circular transcripts (Fig. EV3A). We included host transcripts, such as *7SK*, *GAPDH*, and circHIPK3 for reference.

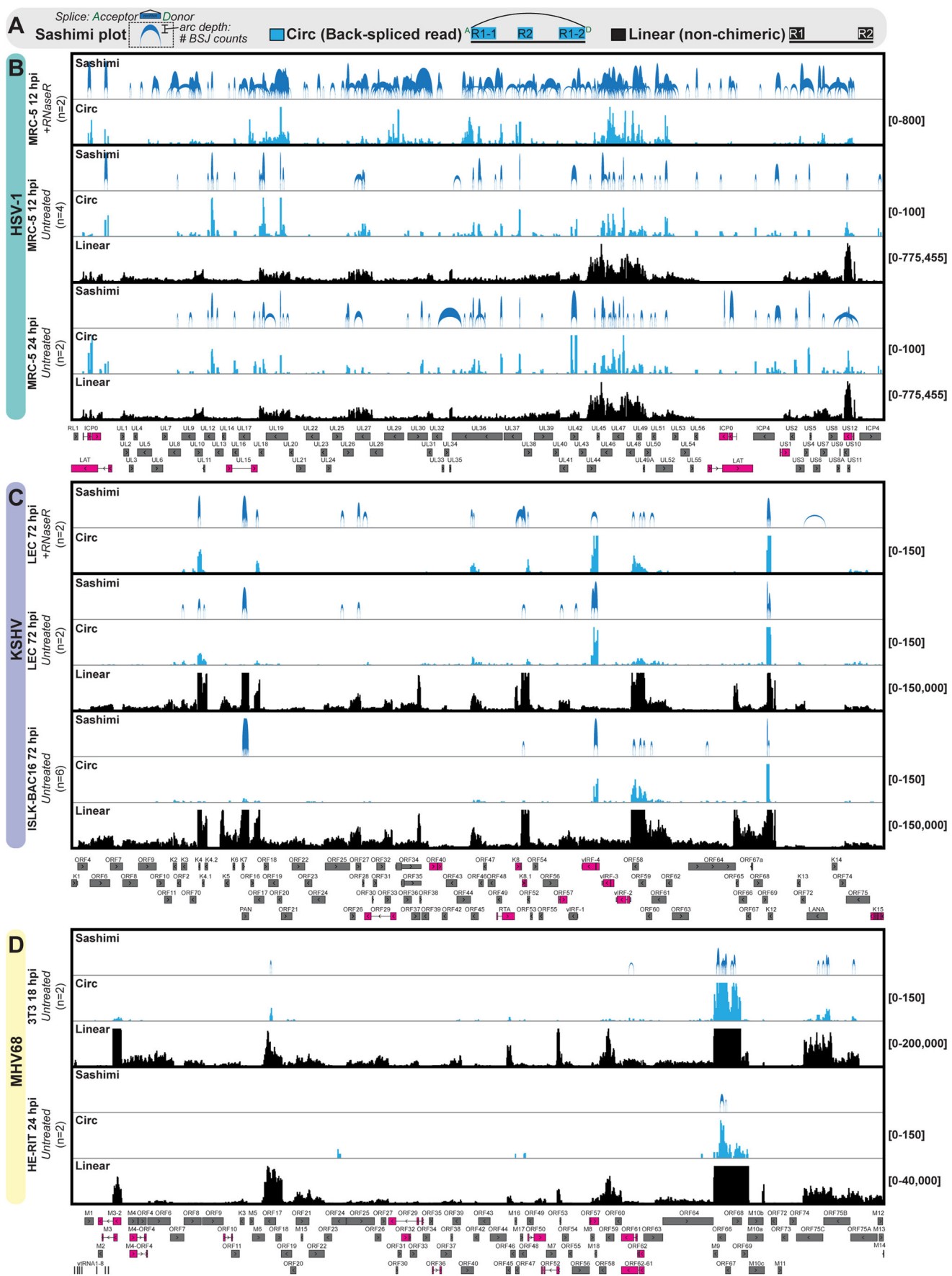

◄ **Figure 2. Profile of high-incidence lytic circRNAs.**

(A) Infographic for data visualization. (B–D) Visualization of high-confidence circRNAs for HSV-1, KSHV, and MHV68 in lytic models described in Fig. 1. If indicated, RNA samples were enriched for circRNAs using RNase R (+ RNase R). Single-exon (gray) and multi-exon (pink) viral genes are shown below. Data Information: RNA-Seq was performed and high-confidence circRNAs were called using CHARLIE. Sashimi plots are limited to circRNA variants identified in at least two biological replicates, arcs are proportional to raw BSJ counts. Blue and black traces include circular (back-spliced reads) and linear (non-chimeric) reads, respectively. Traces are the sum of raw BSJ or linear read values for all biological replicates. Y axis minimum and maximum values are shown on the right.

Viral circRNAs ranged in abundance from $10^3$ to $10^6$ copies per µg total RNA. HSV-1 UL44 and KSHV PAN were the most abundant, reached $10^5$ and $10^6$ copies/µg total RNA, respectively. These levels are similar or greater than circHIPK3, a highly abundant human circRNA with published roles in miRNA sponging (Zheng et al, 2016). We used the anticipated full-length sequence of HSV-1 circLAT, KSHV circK12, and MHV68 circORF66 to predict circRNA–miRNA and miRNA-mRNA interaction partners (Fig. EV3B). This analysis has the caveat that we used the entire sequence between the 5' and 3' BSJ and this may not be true if additional splicing events are present. Downstream phenotypes that may ultimately be controlled by these circRNA–miRNA–mRNA interaction networks included senescence and mTOR signaling (Fig. EV3C). Subsequent work is necessary to follow-up on these in silico predictions. These findings identify back-splicing hotspots within the viral genome, with select lytic species reaching abundance only 100-fold less than the housekeeping transcript, *GAPDH*.

## *Cis*-elements coordinating viral back-splicing events

When visualizing data mapped to the viral genome, we observed many circRNAs derived from single-exon genes (Fig. 2). The lack of relationship between forward and back splicing was particularly notable for HSV-1, with circRNA species tiling the viral genome (Fig. 2B). This is perhaps unsurprising as only five of the ~80 genes are spliced and HSV-1 itself inhibits the host splicing machinery (Hardy and Sandri-Goldin, 1994; Tang et al, 2016). Thus, we investigated flanking *cis*-elements to compare high-confidence host and viral circRNA species (Fig. 3A). First, we determined the 2 bases 5' and 3' to BSJ, as these would be the splice donor–acceptor (DA) sites. As an alternative splicing event, circRNAs are expected to rely on canonical splice DA signals, such as GU-AG (Ashwal-Fluss et al, 2014; Starke et al, 2015). As predicted, 84–97% of host circRNAs used GU-AG as their splice donor–acceptor (Fig. 3B; Appendix Fig. S4A). Of the top ten most highly expressed species for each virus, two HSV-1 circRNA (circICP0-exon2, circUL29), two KSHV circRNA (circvIRF4, circPAN), and zero MHV68 circRNA possess GU-AG/CU-AC as their flanking splice DA (Dataset EV1). When we examined all high-confidence viral circRNAs, we found that only 0.6–7% used the canonical splice DA (Fig. 3B). Similar trends were observed for circRNAs identified in KSHV and MHV68 reactivation models (Appendix Fig. S4A). Analysis of circRNAs detected in RNase R-digested samples corroborated our findings (Appendix Fig. S4A). In summary, >90% of viral circRNAs are not flanked by canonical splice DA elements.

Next, we evaluated GC content of the 100 nt flanking the BSJ (Fig. 3C,D). GC content directly impacts many facets of gene expression including, transcription rates, splicing, and secondary structure. Previous work has found that low GC content is a significant predictor of circRNA hotspots (Gruhl et al, 2021). Consistent with this, we found the GC content of external sequences flanking human and mouse circRNAs was significantly lower than the theoretical average for host genes (Fig. 3C,D). This effect was most striking for human circRNAs with an average decrease of 9% relative to host gene GC content. GC content in KSHV and MHV68 circRNA flanking sequences followed a similar trend, with a decrease of 2.7 and 3.1%, respectively. Viral circRNAs from lytic reactivation models had more drastic decreases in GC content with 7.4% for KSHV and 5.6% for MHV68 (Appendix Fig. S4B). HSV-1 circRNAs were the outlier with a GC-content increase of 2.6%; this increase was relative to the already high GC content of HSV-1 genes (67%). To put this in context, HSV-1 circRNA flanking sequences had a GC content of 70% as compared to host derived circRNAs with a GC content of 36%. All of our findings were consistent when comparing circRNA identified in untreated or RNase R-digested samples (Fig. 3C,D; Appendix Fig. S4B).

To identify novel motifs with the potential to recruit RBPs and coordinate circRNA synthesis, we performed motif enrichment analysis. Supporting our findings in Fig. 3C, we found uracil tracks to be overrepresented in host circRNA flanking sequences (Fig. 3E). This was by far the most enriched motif with *E*-values < 4.3E-62. Whether poly-U truly functions as a motif to coordinate protein binding or is merely indicative of the low GC content in host circRNA flanking regions remains to be explored. Distinct motifs were found for regions flanking viral circRNAs, with the most enriched being HSV-1: GGKUUUUWUD, CAWGGMGCCC, GCGKCGWCGU; KSHV: WUGWUGUUUU, CAGUCMCMKU, CCCMUUUWU; MHV68: ASAGGGUCAG, USKCCACCWC, ARUGUGUUDU. Of these motifs, three contain uracil tracks, similar to host circRNA flanking elements. Five (ANKHD1, FUS, HNRNPH2, HNRNPK, PCBP2) of the 27 predicted RBP interaction partners have been previously identified to enhance circRNA synthesis using an circmCherry-expression screening system (Li et al, 2017b). We referenced published quantitative mass spectrometry (MS) datasets for HSV-1 primary infection (Soh et al, 2020) and KSHV reactivation (Gabaev et al, 2020) to evaluate changes in protein abundance for our list of 27 predicted RBP interactors (Appendix Fig. S5). Lytic infection drives host shut off resulting in a global decrease in host peptides (Gabaev et al, 2020; Soh et al, 2020). Despite this, four (ACO1, HNRNPH2, SART3, SRSF7) and 13 (FUS, G3BP2, HNRNPC, HNRNPCL1, HNRNPH2, HNRNPK, KHDRBS1, MBNL1, PABPC1, PABPC4, SART3, SRSF7, ZC3H) RBPs were upregulated during HSV-1 infection and KSHV lytic reactivation, respectively (Appendix Fig. S5). Whether these upregulated RBPs are responsible for coordinating viral circRNA synthesis remains to be determined. Ultimately our analysis identifies distinct *cis*-elements coordinating viral and host circRNA synthesis and led us to investigate if viral circRNAs are spliceosome products.

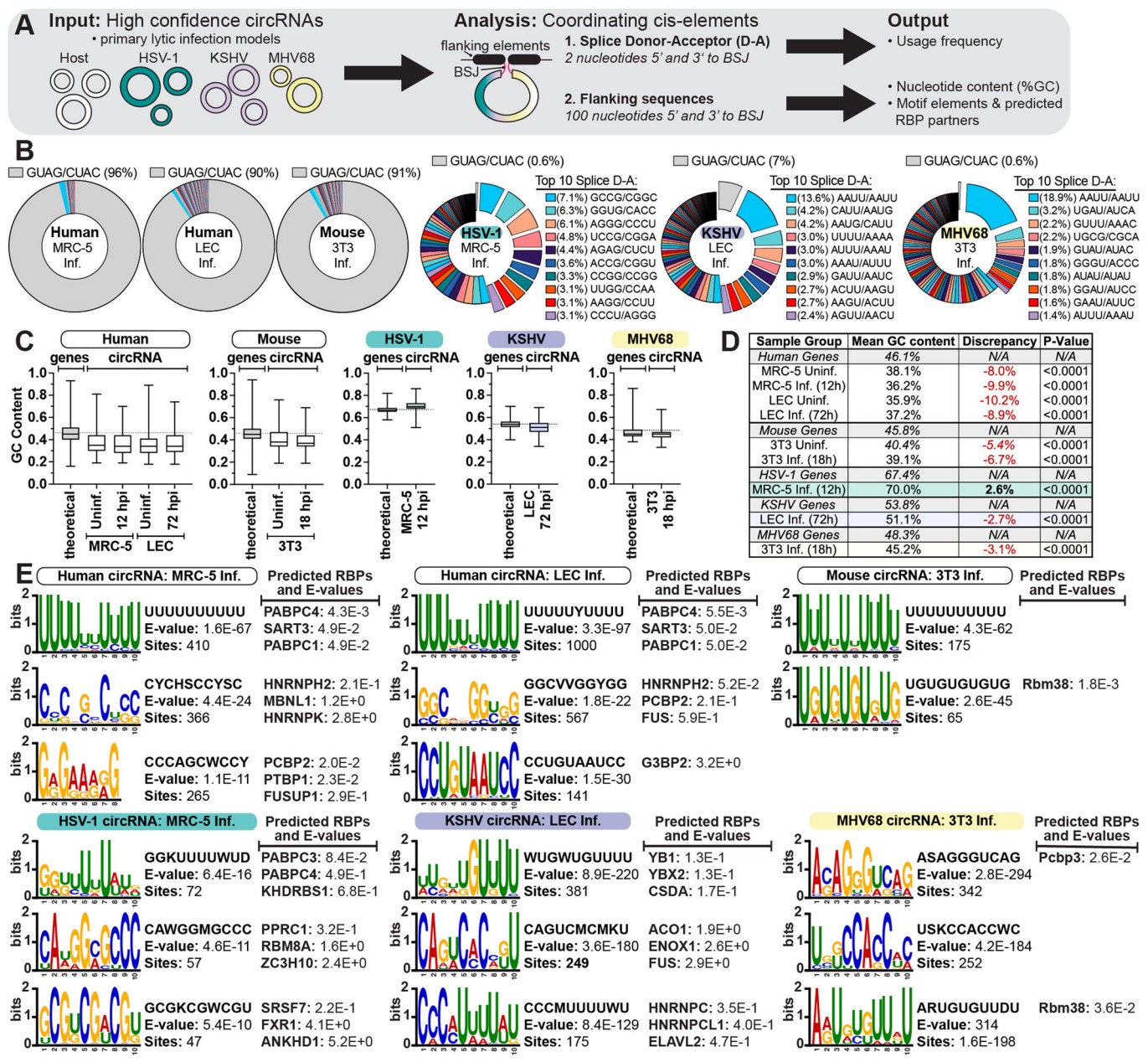

**Figure 3.** *Cis*-elements coordinating viral back-splicing events.

(A) Infographic for BSJ flanking *cis*-element analysis performed on high-confidence viral and host circRNAs identified in primary lytic infection models (infected MRC-5, LEC, 3T3). (B) Splice donor–acceptor frequency for high-confidence BSJ variants, reported as sense and antisense sequences. The percentage which use the canonical splice donor–acceptor are reported above. (C, D) Box and whiskey plots and table summary for GC content of 100 nucleotides (nt) BSJ flanking sequences relative to the theoretical GC content of all genes in an organism. (E) Motif discovery was performed for 100 nt BSJ flanking sequences. The top three motifs were represented as nucleotide consensus plots, with *E*-values, and the number of sites in input sequences to the right. The top three predicted RBP-binding partners for each motif are listed. Data Information: Within box & whisker plots the lower and upper bounds of the box are at the first and third quartiles, respectively, the center line at the median, and the minima and maxima values are indicated by whiskers. For GC-content analysis, Wilcoxon *t* tests were performed relative to the theoretical gene GC content, *P* values are given. Discriminative motif discovery was performed using the MEME algorithm. The top three most significant motifs, by weighted E value, were included. Motif-motif similarity analysis was performed using the TOMTOM algorithm. For predicted RBP-binding partners, all hits had Bonferroni corrected *P* value < 0.05 and only the top three most significant are shown.

## Impact of spliceosome perturbation on viral circRNA synthesis

To perturb the spliceosome, we treated HSV-1 (MRC-5 lytic infection) and KSHV (iSLK-BAC16 lytic reactivation) models with pladienolide B (PB) and isoginkgetin (IGG) which prevent formation of the spliceosome A or B complex, respectively (Fig. 4A). We assessed cell viability and splicing inhibition in our models to determine the appropriate concentration of inhibitor to use to maximize inhibition and minimize off-target effects (Appendix Fig. S6A–D). Based on these assays, we treated our models with 30 μM IGG and 25 nM PB and

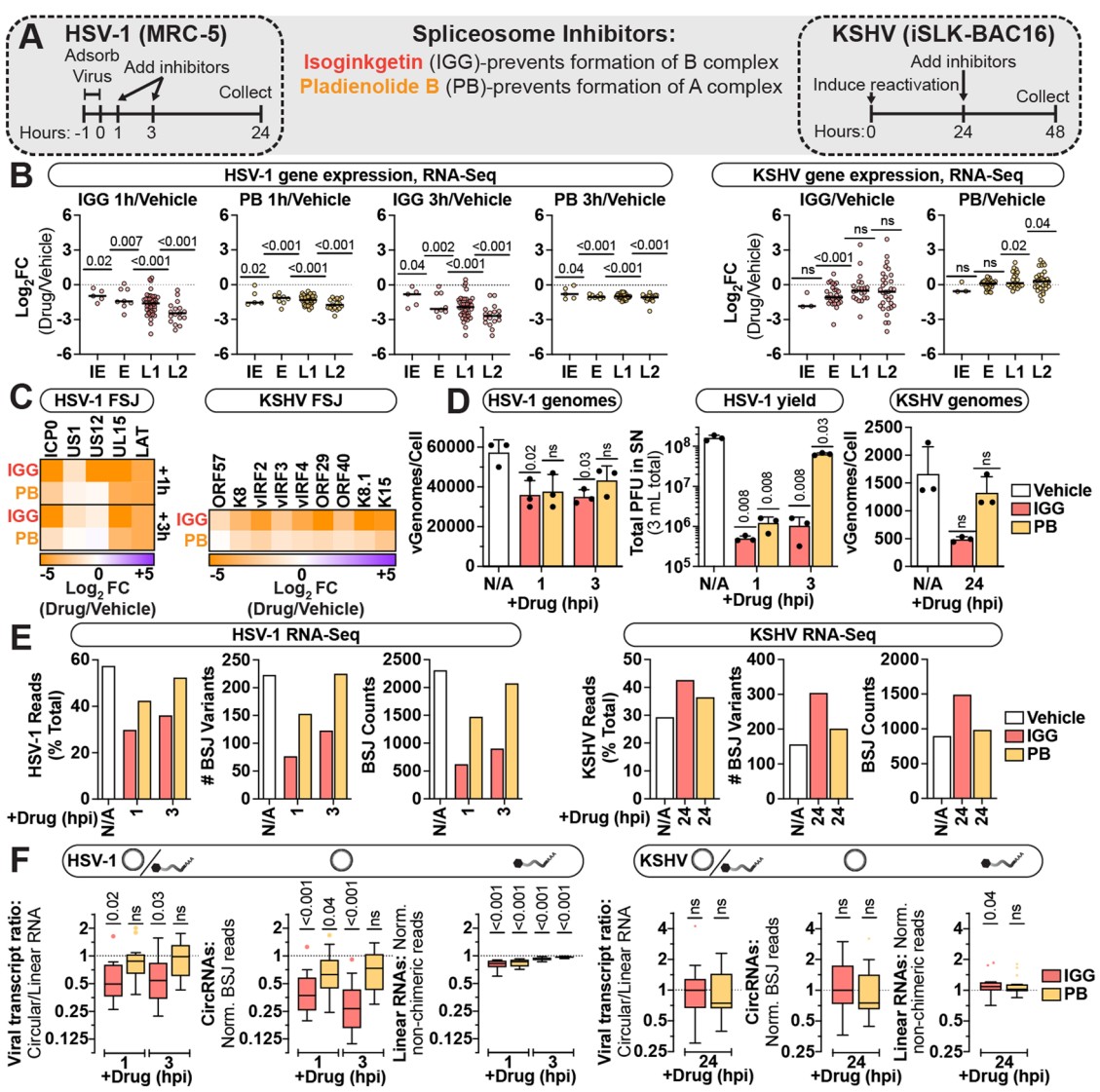

**Figure 4. Impact of spliceosome perturbation on viral circRNA synthesis.**

(A) Experimental infographic. Infection models were treated with spliceosome inhibitors, isoginkgetin (IGG) and pladienolide B (PB). HSV-1-infected fibroblasts were treated with inhibitors at 1 or 3 h post infection and collected at 24 h post infection. iSLK-BAC16 were treated with inhibitors at 24 h after lytic reactivation and collected after 24 h of inhibitor treatment. (B) Viral gene expression (each data point is a gene) clustered by transcriptional class: Immediate early (IE; HSV-1 $n = 5$, KSHV $n = 3$), Early (E; HSV-1 $n = 8$, KSHV $n = 28$), Leaky Late (L1; HSV-1 $n = 41$, KSHV $n = 21$), Late (L2; HSV-1 $n = 17$, KSHV $n = 32$). (C) Expression of spliced viral transcripts, quantified via forward spliced junctions (FSJ). (D) qPCR assessment of genome quantity ($n = 3$), plotted as viral genomes per cell (vGenomes/Cell). HSV-1 yield ($n = 3$) plotted as plaque-forming units (PFU) per mL supernatant (SN). (E) RNA-Seq overview for all viral mapped reads (MR), as percent total reads (% Total). The number of unique BSJ variants and total BSJ counts is reported for high-confidence viral circRNAs. (F) CircRNA quantitation via CHARLIE, graphed as Tukey's box plots. Transcript ratios were determined by plotting gene expression for a given loci, with circular (BSJ-containing reads) over linear (non-chimeric reads). Only circRNA loci with >5 BSJ reads per sample in all conditions were included (HSV-1 $n = 12$, KSHV $n = 18$). Data Information: RNA-Seq data is normalized to ERCC spike-in controls. In viral gene expression graphs, each dot is the average of biological duplicates for a gene, cross-bars are the average. In heatmaps, all data is the average of biological duplicates and plotted as log$_2$FC (Drug/Vehicle). In viral genomes and yield graphs, data points are biological replicates, bar maxima are the average, error bars are standard deviation. For Tukey box plots, the lower and upper bounds of the box are the first and third quartiles, respectively, the center line at the median. The whiskers represent a distance of 1.5 times the IQR relative to the upper or lower quartile; data points are values which did not reside within this range. For viral genomes and yield, paired $t$ tests (Drug-Vehicle) were performed, $P$ values < 0.05 are labeled and not significant (ns) indicates $P$ values ≥ 0.05. For viral circRNA quantitation, Wilcoxon signed-rank tests were performed for each circRNA loci relative to a paired vehicle control, $P$ values < 0.05 are labeled and not significant (ns) indicates $P$ values ≥ 0.05. Source data are available online for this figure.

assessed viral transcription and replication. These drugs successfully inhibited forward splicing for viral transcripts as assessed by qPCR using exon–exon (mature mRNA) or exon–intron (pre-mRNA) specific primers (Appendix Fig. S6E) and quantitation of RNA-Seq forward splice junctions (FSJ) (Fig. 4C). Spliceosome inhibition

resulted in reduced HSV-1 gene expression, genomes and yield (Fig. 4B,D). We expect this is due to decreased levels of the HSV-1 elongation factor (US1 or ICP22), which is a product of forward splicing (Fig. 4C). Spliceosome inhibition had less of an effect on KSHV, with minimal defects in gene expression (Fig. 4B,D).

To determine a global picture of splicing changes we performed RNA-Seq and quantified viral circRNAs using CHARLIE (Fig. 4E,F; Appendix Fig. S7A–D). For HSV-1 there was a decrease in BSJ counts after inhibitor treatment, however this decrease was proportional to global changes in read composition (Fig. 4E). For KSHV there was no decrease in the amount of BSJ reads or variants detected after spliceosome inhibition (Fig. 4E). Similar species of circRNAs were expressed during vehicle or drug treatment (Appendix Fig. S7A–C). We calculated the ratio of circular/linear reads for a subset of circRNAs expressed in vehicle and inhibitor-treated conditions. Using this metric, we can evaluate how perturbation alters the likelihood a given gene gives rise to a circRNA. Inhibitor treatment only caused a significant shift in circ/linear ratios for HSV-1 + IGG 1 or 3 hpi (Fig. 4F). All other conditions did not have a significant change in circRNA synthesis (Fig. 4F). In Appendix Fig. S7D,E, we highlight a subset of HSV-1 circRNAs, one with and one without canonical splice DA elements. The circRNA species which arises from the middle exon of HSV-1 ICP0 is undetectable or decreased after spliceosome inhibition (Appendix Fig. S7D). By comparison, the circRNA species colinear to UL12 is unaffected by spliceosome inhibition (Appendix Fig. S7D). As general transcriptional output from HSV-1 ICP0 was decreased by spliceosome inhibition we checked our Illumina SRS data via a more sensitive method, namely digital droplet PCR. By ddPCR, circICP0 was still detectable after spliceosome inhibition, with a marginal decrease in circ/linear ratios observed after IGG treatment (Appendix Fig. S7E). Echoing our SRS results, the circ/linear ratio for UL12 was unaffected by spliceosome inhibition (Appendix Fig. S7E). To corroborate our KSHV findings we performed siRNA depletion of a core component of the spliceosome, *PRPF8* (Appendix Fig. S8). After *PRPF8* depletion we observed a decrease in host and viral forward splicing (Appendix Fig. S8B). Consistent with prior reports (Liang et al, 2017), host circTNPO3 was more resistant to spliceosome perturbation than its colinear gene, resulting in an increased circ/linear ratios for the loci (Appendix Fig. S8C–E). By comparison, viral circRNAs (circvIRF4, circPAN, circK2) were unaffected by PRPF8 depletion (Appendix Fig. S8C–E). Thus, after spliceosome inhibition and depletion we find KSHV circRNA synthesis unaffected.

During splicing, the intron is excised and forms a lariat species. While the lariat is circular in shape, it differs from true circRNAs as it is created by a phosphodiester bond and has a "tail" branching from the 2'–5' junction (Ruskin et al, 1984). The RNA lariat debranching enzyme (DBR1) subsequently cleaves the phosphodiester bond allowing for exonucleases to degrade the now linear species (Ruskin and Green, 1985). While the intron "tail" largely prevents reverse transcription (RT) from proceeding across lariat 2' to 5' junctions, there is still the potential for these species to be picked up by sequencing and misrepresented as true circRNAs (Suzuki et al, 2006). Thus, we examined the possibility that viral circRNAs may be splicing lariats that are inappropriately identified as circRNAs. We used siRNAs to knockdown *DBR1* preceding lytic HSV-1 and KSHV infection (Fig. EV4). After siRNA treatment we observed a significant depletion in protein and RNA levels of *DBR1* for all conditions (Fig. EV4A–C,E). Consistent with *DBR1* depletion, we detected a >10-fold increase in *DKK1* and *ID1*-derived lariats via branchpoint RT assays (Fig. EV4D,F). As expected, human circZKSCAN1, a well-characterized spliceosome product (Liang and Wilusz, 2014), was unaffected by *DBR1* depletion (Fig. EV4G–L). We then assessed viral

circRNAs, testing those with (HSV-1 circICP0; KSHV circvIRF4) or without (HSV-1 circUL12 and circUL44; KSHV circPAN, circK2, circK12) canonical splice donor–acceptor sites. For all loci, viral circRNA levels and circ/linear ratios were comparable between siDBR1 and siNTC treatment (Fig. EV4G–L). These data increase confidence that our computationally identified viral BSJ are from true circRNA and not RNA lariats. In summary, we find viral circRNA synthesis largely unaffected by conditions which perturb spliceosome or lariat debranching activity.

## Dependence of viral circRNAs on host RNA ligases

As viral circRNAs were refractory to spliceosome perturbation we went on to test alternative *trans*-acting factors—namely RNA ligases. There are two known human RNA ligases, RNA 5'-phosphate and 3'-OH ligase 1 (RLIG1) (Yuan et al, 2023) and RNA 2',3'-cyclic phosphate and 5'-OH ligase (RTCB) (Popow et al, 2011). RLIG1 was only discovered in the last year and its in vivo substrates are largely unknown. RTCB has previously been shown to mediate circularization of tRNA introns within archaea (*H. volcanii*) and metazoa (*D. melanogaster*) (Lu et al, 2015; Salgia et al, 2003). We depleted these RNA ligases preceding lytic HSV-1 and KSHV infection (Fig. 5A). Following siRNA treatment, we observed a depletion at the RNA and protein level for all expected conditions (Appendix Fig. S9). Viral gene expression was minimally impacted ($log_2$ fold change between $+0.5$ to $-0.5$), with the exception of a defect in KSHV early gene expression after RLIG1 depletion (Fig. 5B,C). RNA ligase (*RTCB* or *RLIG1*) depletion caused a twofold decrease in HSV-1 viral yield, whereas KSHV viral yield was increased approximately twofold by RLIG1 depletion (Fig. 5D).

To determine a global picture of splicing changes we performed RNA-Seq and quantified viral circRNAs using CHARLIE (Figs. 5E and EV5). We observed mild changes in total read composition and BSJ abundance (Fig. EV5A). We calculated the ratio of circular/linear reads for a subset of circRNAs expressed in RNA ligase and nontargeting control (NTC) samples. *RLIG1* depletion caused a significant decrease in circ/linear ratios for HSV-1 and KSHV (Fig. 5E). These results were echoed by the reduction in high-confidence circRNA variants identified after *RLIG1* depletion (HSV-1 NTC, 72 BSJ – RLIG1, 54 BSJ; KSHV NTC, 107 BSJ – RLIG1, 73 BSJ) (Fig. EV5B). *RTCB* depletion caused a significant shift in circ/linear ratios for KSHV (Fig. 5E), with a corresponding decrease in BSJ variability (KSHV NTC, 107 BSJ – RTCB, 81 BSJ) (Fig. EV5B). We used ddPCR to investigate loci-specific changes for a subset of viral and host circRNAs, including human ZKSCAN1; HSV-1 circICP0, circLAT, circUL12, circUL44; KSHV circPAN, circK2, circvIRF4 (Fig. 5F; Appendix Fig. S10). To account for general effects on gene expression, we plotted the ratio of circular/linear transcripts to evaluate how perturbation alters the likelihood that a given gene gives rise to a circRNA. The spliceosome-dependent host circRNA (circZKSCAN1) was unaffected by RNA ligase depletion (Appendix Fig. S10). Of the seven viral circRNAs tested, two (circICP0, circLAT) were significantly decreased by *RLIG1* depletion and four (circICP0, circLAT, circPAN, circK2) were decreased by *RTCB* depletion (Fig. 5C,D). HSV-1 circUL12, HSV-1 circUL44, and KSHV circvIRF4 circ/linear ratios were unchanged by RNA ligase depletion (Appendix Fig. S10). These data support RNA ligases as novel *trans*-acting factors in viral circRNA synthesis, with loci-specific dependencies.

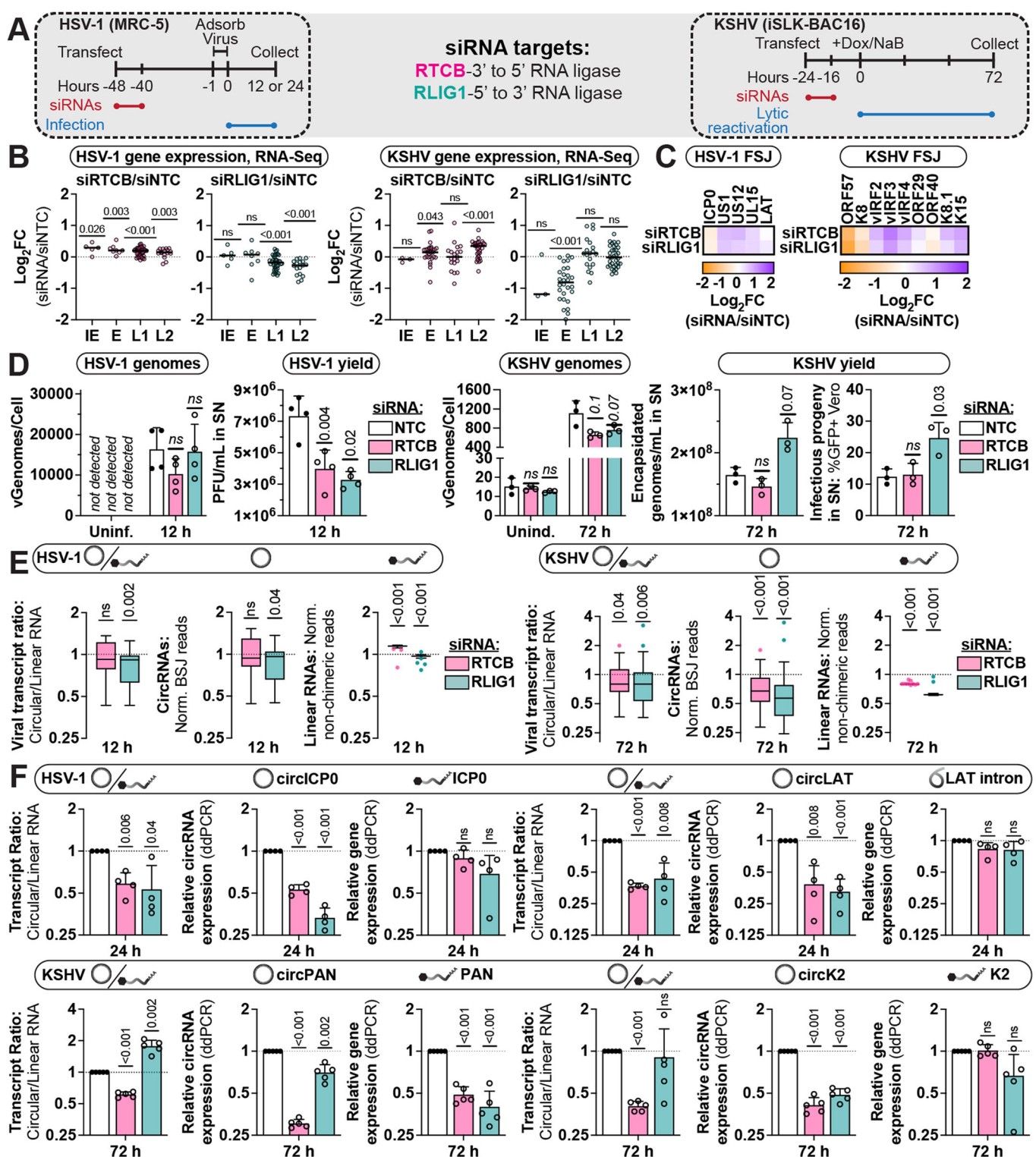

## Impact of key lytic effectors on viral circRNA synthesis

While herpesviruses replicate in the nucleus and rely on the host machinery for transcription, splicing, and RNA export, they express viral proteins which target and modulate these pathways. To investigate how these unique effectors may influence circRNA

biogenesis, we employed knockout viruses and inhibitors that target distinct phases of the lytic life cycle. These included viral transcription factors (HSV-1 ICP4 and ICP22, KSHV ORF24, MHV68 ORF50) and RNA-binding proteins (HSV-1 ICP27, KSHV ORF57). We used cycloheximide (CHX) to block HSV-1 at the phase of immediate early transcription and DNA replication

◀ **Figure 5. Dependence of viral circRNAs on host RNA ligases.**

(A) Experimental infographic. MRC-5 were transfected with siRNAs targeting RNA ligases (*RLIG1, RTCB*) or a nontargeting control (NTC) for 48 h. MRC-5 were infected with HSV-1 for an additional 12 or 24 h (12 or 24 h). iSLK-BAC16 were siRNA transfected for 24 h followed by lytic reactivation for 72 h (72 h). (B) Viral gene expression (each data point is a gene) clustered by transcriptional class: Immediate early (IE; HSV-1 $n = 5$, KSHV $n = 3$), Early (E; HSV-1 $n = 8$, KSHV $n = 28$), Leaky Late (L1; HSV-1 $n = 41$, KSHV $n = 21$), Late (L2; HSV-1 $n = 17$, KSHV $n = 32$). (C) Expression of spliced viral transcripts, quantified via forward spliced junctions (FSJ). (D) qPCR assessment of genome quantity ($n = 3$–4) plotted as viral genomes per cell (vGenomes/Cell). HSV-1 yield ($n = 3$) plotted as plaque-forming units (PFU) per mL supernatant (SN). KSHV progeny ($n = 3$) plotted as protected viral genomes (encapsidated genomes per mL in SN) or infectivity measured via supernatant transfer (%GFP+ Vero). (E) CircRNA quantitation via CHARLIE, graphed as Tukey's box plots. Transcript ratios were determined by plotting gene expression for a given loci, with circular (BSJ-containing reads) over linear (non-chimeric) reads. Only circRNA loci with >5 BSJ reads per sample in all conditions were included (HSV-1 $n = 30$, KSHV $n = 50$). (F) ddPCR quantitation using divergent (circRNA) or convergent (gene) primers ($n = 4$–5). Data Information: RNA-Seq data is normalized to ERCC spike-in controls. In viral gene expression graphs, each dot is the average of biological duplicates for a gene, cross-bars are the average. In heatmaps, all data is the average of biological duplicates and plotted as $\log_2$FC (RNA ligase/NTC). In column bar graphs, data points are biological replicates, bar maxima are the average, error bars are standard deviation. For Tukey box plots, the lower and upper bounds of the box are the first and third quartiles, respectively, the center line at the median. The whiskers represent a distance of 1.5 times the IQR relative to the upper or lower quartile; data points are values which did not reside within this range. For viral genomes, viral yield, and ddPCR assays paired $t$ tests (RNA ligase-NTC) were performed, $P$ values < 0.05 are labeled and not significant (ns) indicates $P$ values ≥0.05. For circRNA quantitation (RNA-Seq), Wilcoxon signed-rank tests were performed for each circRNA loci relative to a paired NTC control, $P$ values < 0.05 are labeled and not significant (ns) indicates $P$ values ≥ 0.05. Source data are available online for this figure.

inhibitors (cidofivir or CDV, phosphonoacetic acid or PAA) to block viruses at the phase of immediate early and early transcription. We quantified viral genome replication and transcription, confirming that our infections echoed previously published phenotypes (Appendix Figs. S11 and S12A,B) (DeLuca et al, 1985; Honess and Roizman, 1974; Jones and Roizman, 1979; McCarthy et al, 1989; Nandakumar and Glaunsinger, 2019; Pavlova et al, 2003; Rice et al, 1995; Ruiz et al, 2019). We performed RNA-Seq and quantified viral circRNAs using CHARLIE (Fig. 6; Appendix Fig. S12C,D). Viral circRNA abundance and diversity echoed total read composition (Fig. 6A,C), with the outlier being KSHV ΔORF57. In the absence of the KSHV RBP, ORF57, there was a drastic decrease in the diversity of BSJ species (Fig. 6C)—with a new, highly abundant circRNA appearing in the PAN locus (Fig. 6D,F).

To account for global transcriptional changes in these infections, we quantified the circ/linear transcript ratio for only actively transcribed regions of the genome. We observed a higher circ/linear ratio for viral genes in the absence of the viral transcription factors HSV-1 ICP4 and KSHV ORF24 (Fig. 6B,D). As the number of circRNA species generated in these conditions is much lower than wild-type, we are uncertain of the significance of this shift. No MHV68 circRNAs were identified in the absence of the viral transcription factor, ORF50, thus circ/linear ratio could not be determined (Appendix Fig. S12C,D). Inhibition of viral DNA replication had minimal impact on circ/linear ratios for HSV-1, KSHV, and MHV68 (Fig. 6A–D; Appendix Fig. S12C,D). We observed no significant change in HSV-1 circRNA synthesis in the absence of the viral RBP (ICP27) or transcription elongation factor (ICP22) (Fig. 6E; Appendix Fig. S13A). We plotted the top five most highly expressed BSJ variants from each condition and compared their levels (Fig. 6E,F). For HSV-1, we noted that the colinear genes which give rise to highly expressed circRNAs echoed the kinetic class licensed in each condition (Appendix Fig. S13A). The shift in profile during HSV-1 infection leads us to suspect no specific viral effector tested was responsible for promoting circRNA biogenesis, and instead that any highly expressed viral gene has the potential to generate a circular isoform. For KSHV, we found only the viral RNA-binding protein, ORF57, significantly altered the viral circRNA profile.

## ORF57 regulation of circRNA synthesis

Next, we investigated the potential role that KSHV ORF57 may have in controlling circRNA biogenesis. As ORF57 is a viral RNA-binding protein (Nekorchuk et al, 2007; Sei et al, 2015), we first assessed if it directly bound circRNAs. We performed eCLIP (enhanced version of the crosslinking and immunoprecipitation) on iSLK-BAC16 reactivated for 24 h by treating with Dox/NaB (Fig. 7A,B). ORF57 immunoprecipitated (IP) or size-matched input (input) samples were sequenced to generate 50 bp single-end reads. Forward and back splice junctions were quantified using STAR or CHARLIE. We observed a general enrichment for viral reads within ORF57 IP samples (Fig. 7A). ORF57 IP resulted in an enrichment for KSHV and host back-spliced transcripts as well as host forward-spliced transcripts (Fig. 7B). Of the 71 viral BSJ enriched after ORF57 IP, 59 were from species colinear with PAN (Appendix Fig. S14A). Host genes with ≥2 BSJ variants enriched in the ORF57 IP were *GLS*, *RMRP*, *MT-RNR2* (Appendix Fig. S14B). These data demonstrate that ORF57 binds a subset of host and viral circRNAs within close proximity (50 nt) of the back-spliced junction.

ORF57 possesses homologs in all human herpesviruses, many of which have been demonstrated to enhance co- and post-transcriptional gene regulatory activity (Malik and Schirmer, 2006; Sandri-Goldin, 2008). We employed Total and Nascent RNA (4sU)-Seq to determine if ORF57 promotes circRNA accumulation by influencing circRNA synthesis or reducing transcript decay (Fig. 7C,D; Appendix Fig. S15). We first evaluated our approach by analyzing linear viral transcripts (Appendix Fig. S15). To compare transcriptional changes, we plotted the $\log_2$ fold change (ΔORF57/WT) for total ($x$ axis) or nascent ($y$ axis) RNA levels. Linear KSHV transcripts at 24 hpi stratified as ORF57 enhanced post-transcriptionally (Appendix Fig. S15B). PAN, ORF6–9, and ORF58-59 were the most downregulated in the absence of ORF57. The notable exceptions to this were K15 and ORF75, which clustered as ORF57 repressed post-transcriptionally. These findings are consistent with a prior study demonstrating that ORF57 prevents premature accumulation of late viral genes by modulating the host RNA decay machinery (Ruiz et al, 2020). This provides confidence that our method can delineate ORF57-mediated co- or post-transcriptional regulatory effects.

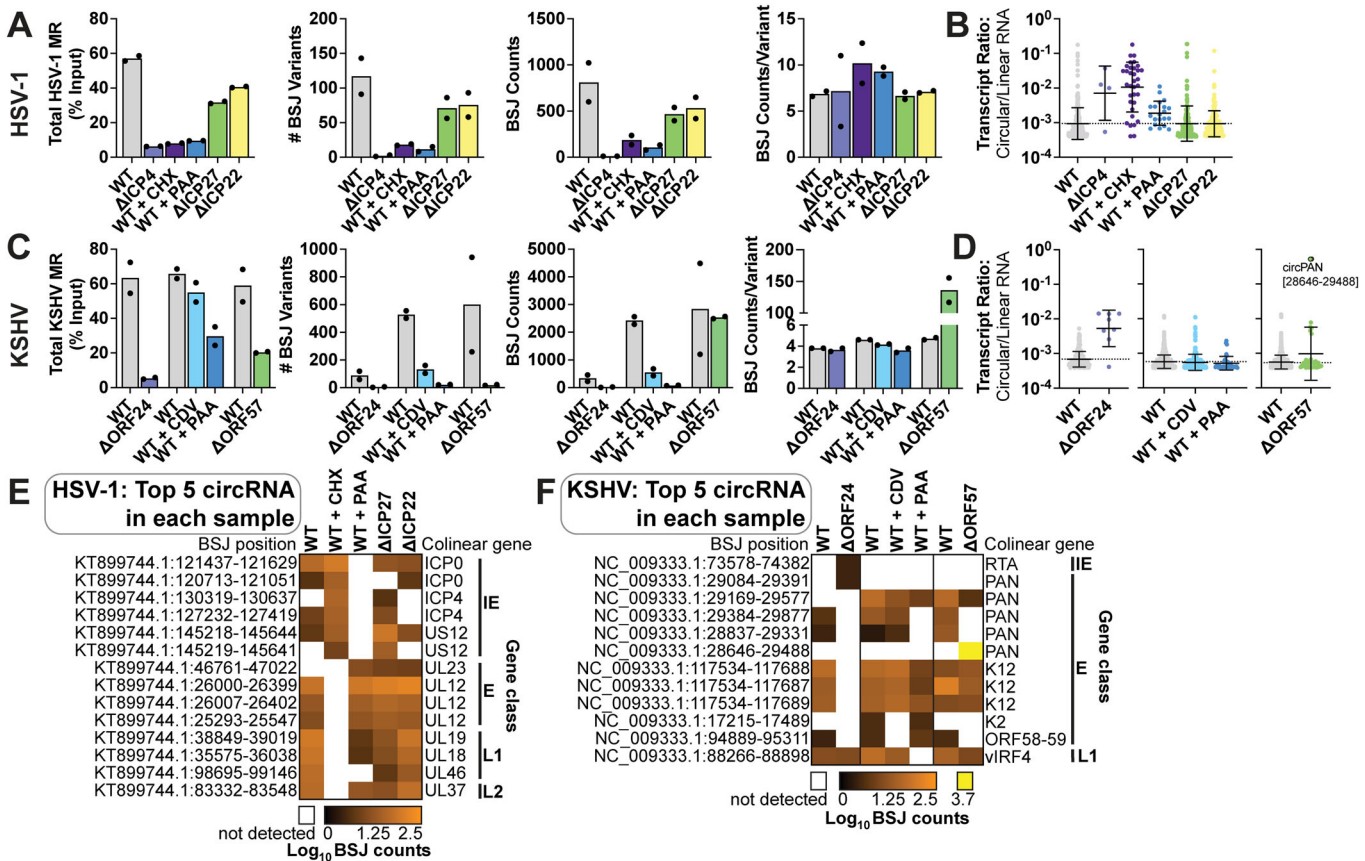

**Figure 6. Impact of key lytic effectors on viral circRNA synthesis.**

(A–F) RNA-Seq data from HSV-1 (MRC-5 cells infected at an MOI of 10 PFU/cell for 12 h) and KSHV (iSLK-BAC16 induced with Dox and NaB for 3 days) infection. Experiments were performed with mutant cells and viruses which lack expression of the viral protein indicated. If indicated, cells were treated with cidofivir (CDV), phosphonoacetic acid (PAA), or cycloheximide (CHX) at 0 h post infection (hpi). (A, C) Sequencing overview for all viral mapped reads (MR), as percent total reads (% Total). The number of unique BSJ variants, total BSJ counts, or BSJ counts/variant is reported for high-confidence viral circRNAs. (B, D) Transcript (circ/linear) ratios for high-confidence viral BSJ variants. (E) The top five most highly expressed circRNAs in each sample were merged to form heatmaps which report $\log_{10}$ BSJ counts for biological duplicates. BSJ positions are reported on the left. Colinear genes and their gene class are reported on the right, with IE (immediate early), E (early), L1 (leaky late), L2 (true late). Data Information: RNA-Sequencing was performed and high-confidence circRNAs were called using CHARLIE. All analysis is relative to a paired wild-type (WT) control. In column bar graphs, data points are biological replicates, and bar maxima are average. In transcript ratio graphs, each dot is the average of biological duplicates for a unique BSJ, cross-bars are the geometric mean and error bars are the geometric standard deviation.

We analyzed our data using CHARLIE to compare changes in host and circRNA levels between KO (ΔORF57) and WT (Fig. 7C,D) KSHV infection. We limited our analysis to uninduced and 24 hpi, to remove any confounding effects from DNA replication defects in KO cells (Appendix Fig. S11A). We observed a greater diversity of viral BSJ variants in both 4sU and total RNA-Seq during WT infection (Fig. 7C). In the absence of ORF57, a novel, highly abundant (>500 raw BSJ reads) circPAN [28646:29488] was expressed; other circPAN species expressed during WT infection were largely absent (Appendix Fig. S16). This novel circPAN variant was at the 5' end of PAN and did not contain the Element for Nuclear Expression (ENE) (Appendix Fig. S16A). circPAN [28646:29488] expression was validated by qPCR using divergent primers and a probe that spans the exact BSJ sequence identified in our RNA-Seq (Appendix Fig. S16B). We examined the expression of other KSHV circRNAs, circK12 and circvIRF4, and saw little difference in expression when comparing KO and WT infection. We support a model in which ORF57 regulates circularization of the PAN transcript, favoring a cluster of many diverse species and

preventing the synthesis of circPAN [28646:29488]. We noted an increase in host circRNAs (BSJ counts) at 24 hpi in our 4sU-Seq data (Fig. 7C). This trend suggests that lytic infection induces the synthesis of host circRNAs. Thus, we evaluated expression changes for host circRNA enriched in our ORF57 eCLIP IP. We found circGLS and circDCBLD2 to be upregulated at 24 hpi, and this upregulation was abrogated in KO cells (Fig. 7D). Host circRNA increase was more pronounced in 4sU-Seq data, suggesting ORF57 promotes synthesis rather than preventing decay. In summary, KSHV ORF57 promotes circRNA synthesis for a subset of viral and host circRNAs during lytic infection.

## Discussion

Herein, we performed comparative circRNA profiling for HSV-1, KSHV, and MHV68 during primary lytic infection, latency, and reactivation. CircRNAs are a novel class of transcripts, with long

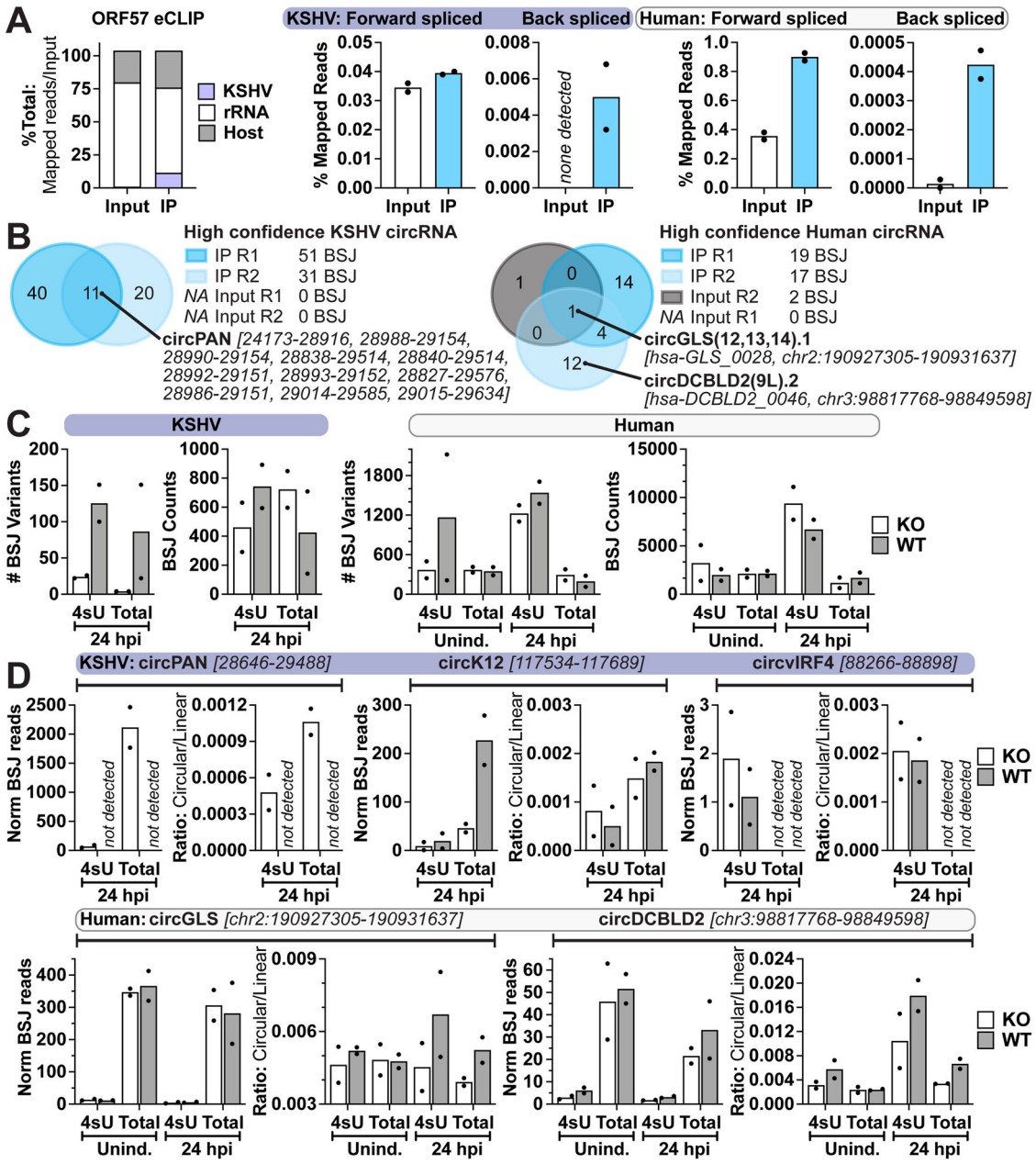

**Figure 7. ORF57 regulation of circRNA synthesis**

(A, B) ORF57 eCLIP (n = 2) was performed on iSLK-BAC16 reactivated for 24 h. "Input" is a paired, size-selected RNA-Seq wherein ORF57 immunoprecipitation (IP) was not performed. (A) Reads mapped to ribosomal RNA (rRNA), human genes (excluding rRNA), or KSHV genes are plotted as a function of total reads. Uniquely mapped reads containing forward or back splice junctions were plotted relative to all reads mapped to their respective genome. (B) Overlapping identity of high-confidence circRNAs called in the dataset. (C, D) Bulk RNA-Seq (Total) and 4sU RNA-Seq (4sU) was performed on ΔORF57 (KO) or iSLK-BAC16 (WT) cells with (24 h) or without (Unind.) reactivation. (C) The number of unique BSJ variants and total BSJ counts is reported for all high-confidence circRNAs. (D) Normalized BSJ counts and circ/linear ratios for select circRNAs. Data Information: eCLIP and RNA-Seq data is the average of biological duplicates. RNA-Seq data was normalized as follows: 4sU-Seq-Reads Per Kilobase per Million mapped reads (RPKM); total relative to ERCC spike-ins. In column bar graphs, data points are biological replicates, and bar maxima are average.

half-lives and low immunogenicity (Kristensen et al, 2019). As alternative splicing products, circRNAs extend transcriptional density without expanding genome space. These qualities certainly seem advantageous to a pathogen in the context of viral infection. We identified thousands of viral circRNAs tiling the viral genome. We generated an annotated resource table with the identity,

position, and sequence of all circRNAs detected in the array of models tested (Dataset EV1). Select viral circRNAs were highly abundant (HSV-1 circUL44, KSHV circPAN) reaching levels within 100-fold of an abundant host housekeeping gene, *GAPDH*. Recent work has shown that viral circRNAs can serve as translational templates (Zhao et al, 2019), be packaged in virions

and extracellular vesicles (Abere et al, 2020), and modulate host cell division and survival (Tagawa et al, 2018; Tagawa et al, 2021a). While abundance does not always equate biological relevance, the high-copy number of these viral circRNAs suggests that they might be prioritized in future studies.

Our profiling identified circRNA species expressed during all phases of the viral life cycle, namely HSV-1 circLAT, KSHV circK12, and MHV68 circORF66. We performed in silico prediction of circRNA–miRNA-mRNA networks for these circRNAs (Fig. EV3B,C). CircRNAs generally function to regulate gene expression, and one mechanism by which this occurs, is miRNA sponging (Kristensen et al, 2019; Tagawa et al, 2021b). In this way, circRNAs repress miRNA-mediated mRNA degradation. Overrepresentation analysis on downstream mRNA targets in the network found that circLAT may modulate cellular senescence by targeting hsa-miR-124-3p. The parental gene, LAT, has previously been shown to influence neuronal senescence and apoptosis, a finding that may establish a link between HSV-1 infection and neurodegenerative diseases (Napoletani et al, 2021; Sivasubramanian et al, 2022). KSHV circK12 was predicted to influence mTOR signaling with AKT3 as a potential downstream mRNA target (Fig. EV3C). A number of KSHV viral proteins have established roles in modulating the PI3K/AKT/mTOR axis (Bhatt and Damania, 2012), as the pathway controls B lymphocyte proliferation and development. One can envision how a non-immunogenic, long-lived, circRNA species with the potential to be expressed during all phases of the life cycle would be an advantageous molecule to also target this axis. These in silico predictions provide testable hypotheses for functional studies of circLAT and circK12 during infection.

CircRNA biogenesis has been largely characterized for model organisms, such as human, mouse, and drosophila. In these models, back splicing is catalyzed by the spliceosome and promoted by RPBs and tandem repeats which mediate interaction of flanking sequences (Ashwal-Fluss et al, 2014; Conn et al, 2015; Jeck et al, 2013; Li et al, 2017b; Starke et al, 2015). As an alternative splicing event, circRNAs rely on canonical splice DA signals, such as GU-AG (Ashwal-Fluss et al, 2014; Starke et al, 2015). Herein, we report that the majority (>90%) of viral circRNAs lack canonical splice donor–acceptor sites (Fig. 3) and do not reside in multi-exon genes (Fig. 2). HSV-1 circRNAs tile the viral genome and appear to be coordinated by high GC content in flanking regions (Figs. 2B and 3C). Depletion of the lariat debranching enzyme (DBR1) had minimal impact on viral circRNAs (Fig. EV4)—lending confidence that CHARLIE successfully delineates bona fide circRNAs from lariat species. Most viral circRNAs were unaffected by spliceosome inhibition or depletion (Fig. 4; Appendix Figs. S7 and S8). These data support a model in which viral back splicing is mediated by machinery other than the major spliceosome, or a model in which viral back splicing is favored over forward splicing when spliceosome activity is limited. The latter model has previously been proposed for circRNAs expressed from the host genome (Liang et al, 2017). As viral back splicing occurs independent of the canonical splice donor–acceptor (GU-AG), we support the first hypothesis for viral circRNA biogenesis. This is of special interest for HSV-1 which only expresses five spliced mRNAs and dysregulates the spliceosome (Hardy and Sandri-Goldin, 1994; Tang et al, 2016).

A spliceosome-independent mechanism for circRNA synthesis was identified in archaea and metazoa (Lu et al, 2015; Salgia et al, 2003). Here, the tRNA splicing machinery creates circularized

tRNA introns via endonuclease cleavage and RTCB catalyzed 3' to 5' RNA ligation. tRNA introns are not flanked by splice DA signals, instead they are recognized by a secondary structure element, called a bulge-helix-bulge motif (Fabbri et al, 1998). This mechanism aligns with our observation that viral BSJs lack a consensus flanking cis-element (Fig. 3B; Appendix Fig. S4A). We would argue that viral transcripts draw a striking parallel to premature-tRNA in that they are short (0.5–1.5 kbp), usually single exon (lack splice donor–acceptor sites), and highly structured (high GC content or repeat elements). The potential for RNA ligase-dependent circRNA synthesis in humans has not previously been explored; although the tRNA splicing machinery is highly conserved, and humans express a homolog of RTCB and the TSEN complex. In addition, a human 5' to 3' RNA ligase (RLIG1, previously called C12orf29) was recently discovered (Yuan et al, 2023). We tested the potential contribution of these RNA ligases to viral circRNA synthesis using loss-of-function models. RLIG1 depletion caused a significant decrease in circ/linear ratios and BSJ variety for HSV-1 and KSHV. RTCB depletion caused a similar shift in the context of KSHV back splicing. Quantification of key instances of viral back splicing, found HSV-1 circICP0, HSV-1 circLAT, KSHV circPAN, KSHV circK2, circvIRF4 to be negatively impacted by RNA ligase depletion (Fig. 5F). Of these loci, circICP0 and circLAT were measured across multiple perturbations including spliceosome inhibition, lariat debranching enzyme depletion, and RNA ligase depletion (Appendix Fig. S7; Figs. EV4 and 5). The only condition to negatively impact circularization was RNA ligase depletion (Fig. 5F). These data support RNA ligases as novel trans-acting factors in viral circRNA synthesis, with loci-specific dependencies. Interestingly, RNA ligase depletion had a negative impact on HSV-1 replication (Fig. 5D). This is at odds with the minimal change in gene expression or replication (Fig. 5B–D) and may support a proviral role for RNA ligase-dependent circRNAs during infection.

CircRNA maturation is frequently mediated by host RNA-binding proteins, which bind adjacent to the 5' and 3' back splice junction and bridge the gap. In line with this, we tested the potential contribution of the KSHV-encoded RBP (ORF57) in promoting viral back splicing. We used complimentary techniques (eCLIP, 4sU-, and Total RNA-Seq) and determined that ORF57 directly bound near back splice junctions in circPAN and shifted the circRNA species expressed from this locus (Fig. 7). ORF57 dependent regulation of PAN may allow for modal expression, such that select species are expressed in abortive vs. replicative lytic infection. This may suggest the ORF57-suppressed species plays a role during abortive lytic infection. Alternatively, ORF57-enhanced species may be antiviral and the virus must suppress their expression until viral infection has progressed through key lytic check points (early gene expression, DNA replication). Further study of these circPAN modalities is necessary to evaluate their contributions during infection contexts. In addition, we found instances of host circRNAs bound by ORF57 (circGLS, circDCBLD2) with increased nascent levels only in the presence of ORF57. By comparing 4sU and Total RNA-Seq, we concluded that ORF57's function is independent of its traditionally published role in modulating RNA decay. Whether ORF57 directly impacts circRNA synthesis through changing the rate of parental gene expression or directly impacting back-splicing rates is unclear. A recent publication demonstrated that ORF57 enhanced the accumulation of the host circRNA, circHIPK3 (Harper et al, 2022). Our study complements this, extending the observation to additional viral and host circRNAs.

Herein, we find that lytic infection results in the synthesis of abundant and diverse viral circRNA species; with a subset expressed in latency and during lytic reactivation. Comparative profiling of host and viral circRNAs after perturbation of key factors—including the spliceosome, lariat debranching enzyme, RNA ligases, viral transcriptional activators, and RNA-binding proteins—identified divergent phenotypes and suggests there are distinct mechanisms of synthesis. As viral circRNAs have only been studied in the last 7 years, our work has the potential to identify new factors which contribute to herpesvirus replication, persistence, and tumorigenesis. This in turn may aid in the identification of novel clinical biomarkers, drug targets, and the development of antiviral compounds.

# Methods

### Reagents and tools table

| Reagent/resource | Reference or source | Identifier or catalog number |
|---|---|---|
| **Experimental models** | | |
| Vero | ATCC | CCL-81 |
| E5 (Vero-based, ICP4 complementing) | DeLuca and Schaffer, 1987 | N/A |
| E11 (Vero-based, ICP4 and ICP27 complementing) | Samaniego et al, 1995 | N/A |
| MRC-5 | ATCC | CCL-171 |
| A20 HE-RIT | Santana et al 2017 | N/A |
| iSLK-BAC16 | Brulois et al 2012 | N/A |
| iSLK-BAC16ΔORF57 | Ruiz et al, 2019; Ruiz et al, 2019 | N/A |
| iSLK-BAC16 ORF24stop | Nandakumar and Glaunsinger, 2019; Nandakumar and Glaunsinger, 2019 | N/A |
| HDLEC | PromoCell | C-12216 |
| HUVEC | Lonza | C2519A |
| NIH 3T3 | ATCC | CRL-1658 |
| NIH 3T12 | ATCC | CCL-164 |
| CS-RTA4 (NIH 3T12-based, RTA complementing) | Li et al, 2017a | N/A |
| 5dl1.2 (strain KOS, ICP27-deletion) | McCarthy et al, 1989 | N/A |
| n199 (strain KOS, ICP22-nonsense) | Rice et al, 1995 | N/A |
| n12 virus (strain KOS, ICP4-nonsense) | DeLuca et al, 1985 | N/A |
| KOS | Smith, 1964 | N/A |
| ORF50stop | Pavlova et al, 2003 | N/A |
| H2B-YFP | Collins et al, 2009 | N/A |
| **Antibodies** | | |
| VCL | Abcam | ab129002 |
| DBR1 | ProteinTech | 16019-1-AP |
| RTCB | ProteinTech | 19809-1-AP |
| RLIG1 (C12orf29) | ThermoFisher | PA5-37983 |
| ICP8 | Abcam | ab20194 |
| ORF57 | Santa Cruz Biotech | sc-135746 |
| GAPDH | Abcam | ab8245 |
| Goat anti-Rabbit IRDye 680 | LI-COR | 926-32221 |
| Goat anti-Rabbit IRDye 800 | LI-COR | 925-32211 |

| Reagent/resource | Reference or source | Identifier or catalog number |
|---|---|---|
| Goat anti-Mouse IRDye 680 | LI-COR | 926-32220 |
| Goat anti-Mouse IRDye 800 | LI-COR | 926-32210 |
| Donkey anti-Goat IRDye 800 | LI-COR | 926-32214 |
| **Oligonucleotides and other sequence-based reagents** | | |
| TaqMan gene expression assay-DBR1 | ThermoFisher | Hs01113907_m1 |
| TaqMan gene expression assay-RLIG1 | ThermoFisher | Hs01594970_m1 |
| TaqMan gene expression assay-PRPF8 | ThermoFisher | Hs00197615_m1 |
| TaqMan gene expression assay-18S | ThermoFisher | Hs99999901_s1 |
| TaqMan gene expression assay-RPS13 | ThermoFisher | Hs01011487_g1 |
| DNAJB1_ PCR_F | IDT | GAACCAAAATCACTTTCCCCAAGGAAGG |
| DNAJB1_PCR_R | IDT | AATGAGGTCCCCACGTTTCTCGGGTGT |
| hu_GAPDH_F | IDT | CAGAACATCATCCCTGCCTCTACT |
| hu_GAPDH_R | IDT | GCCGAGCTTCCCGTTCA |
| hu_RPS13_F | IDT | TCGGCTTTACCCTATCGACGCAG |
| hu_RPS13_R | IDT | ACGTACTTGTGCAACACCATGTGA |
| hu_PreDNAJB1_F | IDT | GGCCTGATGGGTCTTATCTATGG |
| hu_PreDNAJB1_R | IDT | TTAGATGGAAGCTGGCTCAAGAG |
| hu_PreBRD2_F | IDT | AGGTAATGTCACAGGATGGGAAGT |
| hu_PreBRD2_R | IDT | CCCTGCTGCCTTTCTCTAACC |
| hu_MatureDNAJB1_F | IDT | GAACCAAAATCACTTTCCCCAAGGAAGG |
| hu_MatureDNAJB1_R | IDT | AATGAGGTCCCCACGTTTCTCGGGTGT |
| hu_MatureBRD2_F | IDT | CAAAATTATAAAACAGCCTATGGACATG |
| hu_MatureBRD2_R | IDT | TTTTCCAGCGTTTGTGCCATTAGGA |
| hu_DKK1_mRNA_F | IDT | CCTTGGATGGGTATTCCAGA |
| hu_DKK1_mRNA_R | IDT | CCTGAGGCACAGTCTGATGA |
| hu_DKK1_lariat_BP | IDT | GCCCGACCCCTCTCACTGAG |
| hu_DKK1_lariat_F | IDT | GAGGGAGTAGAACGTGCTGA |
| hu_DKK1_lariat_R | IDT | GCCCGACCCCTCTCACTGAG |
| hu_ID1_mRNA_F | IDT | AAACGTGCTGCTCTACGACA |
| hu_ID1_mRNA_R | IDT | CTCCAACTGAAGGTCCCTGA |
| hu_ID1_lariat_BP | IDT | GGTCGGATCTGGATCTCACTTGG |
| hu_ID1_lariat_F | IDT | CCACTTCCGTCCCATCCTT |
| hu_ID1_lariat_R | IDT | GGTCGGATCTGGATCTCACTTGG |
| hu_TNPO3_mRNA_F | IDT | ACATTGCAGCTCGTGTACCA |
| hu_TNPO3_mRNA_R | IDT | AGCATGACTCCACATCCTGC |
| hu_ZKSCAN1_mRNA_F | IDT | ATGAGGGTAGTCCCAGAGACC |
| hu_ZKSCAN1_mRNA_R | IDT | CTGAGATTCCTCCGAGCCAG |
| hu_18S_F | IDT | GTAACCCGTTGAACCCCATT |
| hu_18S_R | IDT | CCATCCAATCGGTAGTAGCG |
| hu_7SK_F | IDT | TAAGAGCTCGGATGTGAGGGCGATCTG |
| hu_7SK_R | IDT | CGAATTCGGAGCGGTGAGGGAGGAAG |
| mmuGAPDH_F | IDT | CTGACGTGCCGCCTGGAGAAAC |
| mmuGAPDH_R | IDT | CCCGGCATCGAAGGTGGAAGAGT |
| hsv_ICP0_F | IDT | CCCACTATCAGGTACACCAGCTT |
| hsv_ICP0_R | IDT | CTGCGCTGCGACACCTT |
| hsv_UL12_F | IDT | TCCCTGGATATTCTCGTCTGTC |
| hsv_UL12_R | IDT | CCGGCATTTGACCTCGTAAA |
| hsv_UL23_F | IDT | ACCCGCTTAACAGCGTCAACA |
| hsv_UL23_R | IDT | CCAAAGAGGTGCGGGAGTTT |
| hsv_UL29_F | IDT | CGAACTTGCGGGTGCGGTCAAA |

| Reagent/resource | Reference or source | Identifier or catalog number |
| --- | --- | --- |
| hsv_UL29_R | IDT | CATGGTCGTGTTGGGGTTGAGCATC |
| hsv_UL42_F | IDT | GTCCCGCCGCTCCAGAC |
| hsv_UL42_R | IDT | CTTGCTTCTCCGGTCGGG |
| hsv_UL44_F | IDT | GTGACGTTTGCCTGGTTCCTGG |
| hsv_UL44_R | IDT | GCACGACTCCTGGGCCGTAACG |
| hsv_PreICP0_F | IDT | GATCCAAAGGACGGACCCAG |
| hsv_PreICP0_R | IDT | GATTTCCCGCGTCAATCAGC |
| hsv_MatureICP0_F | IDT | GCGAGTACCCGCCGGCCTGA |
| hsv_MatureICP0_R | IDT | CTCGAACAGTTCCGTGTCC |
| hsv_PreUL15_F | IDT | CCCACCCACATACACACACA |
| hsv_PreUL15_R | IDT | CTCCTCAAGCGATCCCGAAT |
| hsv_MatureUL15_F | IDT | CCCGAGTGGACCACGTTAAA |
| hsv_MatureUL15_R | IDT | CTCGTCGACAAAGAGCAGGT |
| hsv_LAT_3'exon_F | IDT | CGCCTTCCCGAAGAAACTCA |
| hsv_LAT_3'exon_R | IDT | CGCTCAATGAACCCGCATT |
| hsv_LAT_Intron_F | IDT | TGTGTGGTGCCCGTGTCTT |
| hsv_LAT_Intron_R | IDT | CCAGCCAATCCGTGTCGG |
| kshv_PAN_F | IDT | GCTCGCTGCTTGCCTTCTT |
| kshv_PAN_R | IDT | CCAAAAGCGACGCAATCAA |
| kshv_ORF6_F | IDT | CTGCCATAGGAGGGATGTTTG |
| kshv_ORF6_R | IDT | CCATGAGCATTGCTCTGGCT |
| kshv_K2_F | IDT | ACCCTTGCAGATGCCGG |
| kshv_K2_R | IDT | GGATGCTATGGGTGATCGATG |
| kshv_K8.1_F | IDT | CCGTCGGTGTGTAGGGATAAAG |
| kshv_K8.1_R | IDT | GTCGTTGTAGTGGTGGCAGAAA |
| kshv_K12_F | IDT | ACCGAGTGCTTTAATGCGGA |
| kshv_K12_R | IDT | AAGCACAATCACGGTTGCAC |
| kshv_vIRF4_F | IDT | CCCAACAGGCCAGCTACATAA |
| kshv_vIRF4_R | IDT | CTTCGTGGAACTCTGAGACGC |
| kshv_PreORF57_F | IDT | TCCCATTTCTAACGTATCGTGCT |
| kshv_PreORF57_R | IDT | CTGTAATATCAAACGCGATAAATGAG |
| kshv_MatureORF57_F | IDT | CAAGCAATGATAGACATGGACATT |
| kshv_MatureORF57_R | IDT | GTCCCTCGATTCGTCAAACT |
| mhv_ORF6_F | IDT | AGGGACAGATTTCCTCAGGTGC |
| mhv_ORF6_R | IDT | CTGGCGTGGAAGCTGTTACC |
| mhv_ORF26_F | IDT | ACTATCTGAGGAGGTGCAC |
| mhv_ORF26_R | IDT | TTTTCCCCTGGGTCAACAC |
| mhv_ORF50_F | IDT | GGCCGCAGACATTTAATGAC |
| mhv_ORF50_R | IDT | GCCTCAACTTCTCTGGATATGCC |
| mhv_LANA_F | IDT | TGTGTGCCAGAAGCTTGTGT |
| mhv_LANA_R | IDT | GCCTTATTTTCCCTTACCAG |
| hu_circRELL1_F | IDT | ATGTCTGTTAGTGGGGCTGA |
| hu_circRELL1_R | IDT | TATCTGCTACCATCGCCTTT |
| hu_circTNPO3_F | IDT | TCGTTCCTTACGAATTGGAG |
| hu_circTNPO3_R | IDT | CTGCCGGATCTGTAACAACT |
| hu_circZKSCAN1_F | IDT | CCTCGAGCTTTGACCTTCATCACG |
| hu_circZKSCAN1_R | IDT | CTCACCTTTATGTCCTGGGAGGT |
| hu_circHIPK3_F | IDT | GTCGGCCAGTCATGTATCAA |
| hu_circHIPK3_R | IDT | TGGAATACACAACTGCTTGGC |
| mmu_circSTAU2_F | IDT | TTCCGTATCCCTCACAGCTC |
| mmu_circSTAU2_R | IDT | CAGGCCACTAGATCCAAAGC |
| mmu_circZFP609_F | IDT | AGGAAGGGGAGAATGAGTGC |
| mmu_circZFP609_R | IDT | TGCCTCCTTGGTCAGAACAT |
| hsv_circICP0-Exon2_F | IDT | ACCACGGACGAGGATGA |
| hsv_circICP0-Exon2_R | IDT | CAGGGAAACACCCAGACATC |
| hsv_circLAT_F | IDT | GACGGGTAAGTAACAGAGTCTAAC |
| hsv_circLAT_R | IDT | ACTATGTTCCTGTTTCTGTCTCC |
| hsv_circUL12_F | IDT | GCCGACCACATAGAGTCAAG |
| hsv_circUL12_R | IDT | CTCACGACCGCATCCAC |
| hsv_circUL19_F | IDT | CCGCGTCAACGTTCATCA |
| hsv_circUL19_R | IDT | CCTGGTCTCTGGAACACG |
| hsv_circUL29_F | IDT | ATCAGGACCTGGCCCTGAG |
| hsv_circUL29_R | IDT | GGAGGCGCATGAGCGTC |
| hsv_circUL42_F | IDT | CGATCCGGAAGACCTCGAT |
| hsv_circUL42_R | IDT | CCTGTAGGATTCCATTAAGTTCGG |
| hsv_circUL44_F | IDT | CTGGTGACTGCCGTGGTG |
| hsv_circUL44_R | IDT | GAGCGGCAGGTGATCGAG |
| kshv_circK2a,b_F | IDT | ATCCCGACGTGACTCCTGA |
| kshv_circK2a_R | IDT | AGAAGCTCCATGACGTCCAC |
| kshv_circK2b_R | IDT | GCCATCGGCGAGCTTTTTAA |
| kshv_circK5_F | IDT | ACCTGTGTAAATTTCGGTGTGGTTT |
| kshv_circK5_R | IDT | GTCCACTTTTCCCCGGCTAT |
| kshv_circPANa_F | IDT | ACCAGACGGCAAGGTTTTTA |
| kshv_circPANa_R | IDT | TCGTTAGTCAACCTAGCAAAACA |
| kshv_circPANb,c_F | IDT | GCCCGATTTACACTCAATCCG |
| kshv_circPANb_R | IDT | CGATTTGAATGACATAGGCGACA |
| kshv_circPANc_R | IDT | TGGTGCGTTGTGAAGCATTT |
| kshv_circPAN_28646-29488_F | IDT | CTTCAAGCTGACCCTTAAT |
| kshv_circPAN_28646-29488_R | IDT | CAGGCTAGTGCTGTAAT |
| kshv_circPAN_28646-29488_Probe | IDT | /56-FAM/TAAGCAAGT/ZEN/CGATTTGAATATTTGGTTTCC/3IABKFQ/ |
| kshv_circORF61_F | IDT | GGAAGAAGCTCATGGACTGG |
| kshv_circORF61_R | IDT | GACCTAAAAACCCGGAGGAG |
| kshv_circvIRF4_F | IDT | CTCCGTGTGGATACCAGTGA |
| kshv_circvIRF4_R | IDT | TGGTTCCACGCAACAGTCT |
| kshv_circK12_F | IDT | CATGGCAGTACATTGCAGCG |
| kshv_circK12_R | IDT | CCGAAGTCAGTGCCACAATT |
| mhv_circORF6_F | IDT | CTGCTGGTCATGAGATAGGAAATAAACT |
| mhv_circORF6_R | IDT | GCTCATGCAGAGAAGATCCAGC |
| mhv_circORF8_F | IDT | CGTAGAGATGCGTGTGGCAC |
| mhv_circORF8_R | IDT | GAGGCCTAACCACTAGTTCAGTCAC |
| mhv_circORF66a_F | IDT | ATCTATTGTTCATATTGTGGGGCG |
| mhv_circORF66a_R | IDT | CAACGCACTATTCGCCACC |
| mhv_circORF66b_F | IDT | GTATGTTTTATTGCAGAGATCAAAAGGAG |
| mhv_circORF66b_R | IDT | GATCTAGGGCCTTGAGGAAGG |
| mhv_circM7_F | IDT | GTAATACCCTCTACAGTCTCGGCC |
| mhv_circM7_R | IDT | GACCTGGGTCTTCTTTTCTTGGT |
| **Chemicals, enzymes, and other reagents** | | |
| JQ1 | Cayman Chemical | CAS: 1268524-70-4 |
| Direct-zol RNA miniprep kit | Zymo | R2053 |
| ReverTra Ace qPCR RT master mix | Toyobo | FSQ-101 |
| Thunderbird Next SYBR qPCR mix | Toyobo | QPX-201 |
| TruSeq Stranded Total RNA Ribo-Zero Gold | Illumina | RS-122-2303 |
| Stranded Total RNA Prep with Ribo-Zero Plus | Illumina | 20040525 |
| ERCC spike-in controls | ThermoFisher | 4456740 |
| **Software** | | |

| Reagent/resource | Reference or source | Identifier or catalog number |
|---|---|---|
| Circrnas in Host And viRuses anaLysis pIpEline (CHARLIE) | https://github.com/CCBR/CHARLIE | N/A |
| Cutadapt v.4.4 | Martin, 2011 | N/A |
| STAR v.2.7.6a | Dobin et al, 2012 | N/A |
| BWA v.0.7.17 | Li and Durbin, 2009 | N/A |
| deepTools2 | Ramírez et al, 2016 | N/A |
| Integrative Genomics Viewer (IGV) | Robinson et al, 2011 | N/A |
| geecee | Afgan et al, 2018 | N/A |
| MEME suite v5.5.4 | Bailey et al, 2015 | N/A |
| Tomtom v5.5.4 | Gupta et al, 2007 | N/A |
| CIRCexplorer2 v.2.3.8 | Zhang et al, 2016 | N/A |
| CIRI2 | Gao et al, 2017 | N/A |
| DCC | Cheng et al, 2015 | N/A |
| circRNAFinder | Westholm et al, 2014 | N/A |
| find_circ | Memczak et al, 2013 | N/A |
| RNAHybrid | Rehmsmeier et al, 2004 | N/A |
| Image Studio | LI-COR v.5.2.5 | N/A |
| Other | | |
| Illumina NextSeq 550 | Illumina | N/A |
| Illumina NovaSeq SP | Illumina | N/A |
| StepOnePlus real-time PCR system | ThermoFisher | N/A |

## Methods and protocols

### Cells and viruses

Vero and Vero-based HSV-1 complementing cell lines (E5, E11) were maintained in DMEM (Gibco #11965-092) supplemented with 5% fetal bovine serum (FBS), 1 mM sodium pyruvate, 2 mM L-glutamine, 100 units/mL penicillin–streptomycin. MRC-5 were maintained in DMEM (Gibco #11965-092) supplemented with 10% FBS, 1 mM sodium pyruvate, 2 mM L-glutamine, 100 units/mL penicillin–streptomycin. NIH 3T3 and NIH 3T12 were maintained in DMEM (Corning # 10-017-CV) supplemented with 8% FBS, 2 mM L-glutamine, 100 units/mL penicillin–streptomycin. A20 HE-RIT B cells harbor a recombinant MHV68 expressing a Hygromycin-eGFP cassette (Forrest and Speck, 2008), and these cells express MHV68 RTA under the control of a doxycycline-inducible promoter (Santana et al, 2017). A20 HE-RIT were maintained in RPMI supplemented with 10% FBS, 100 units/mL penicillin–streptomycin, 2 mM L-glutamine, 50 μM beta-mercaptoethanol, 300 μg/ml hygromycin B, 300 μg/mL gentamicin, and 2 μg/mL puromycin. iSLK-BAC16 renal carcinoma cells harbor a recombinant KSHV expressing a Hygromycin-eGFP cassette, and these cells express KSHV RTA under the control of a doxycycline-inducible promoter (Brulois et al, 2012). iSLK-BAC16 (WT, ΔORF24, ΔORF57) were maintained in DMEM (Gibco #11965-092) supplemented with 10% Tet-Approved FBS, 50 μg/mL hygromycin B, 0.1 mg/mL gentamicin, 1 μg/mL puromycin, 100 units/mL penicillin–streptomycin. HDLEC (LEC) and HUVEC (VEC) were maintained in EBMTM-2 Basal Medium (Lonza #CC-3156) supplemented with EGMTM-2 SingleQuots Supplements (Lonza #CC-4176). We thank Neal DeLuca (n12, 5dl1.2, n199, E5, E11) and Britt Glaunsinger (iSLK-BAC16, iSLK-BAC16 ΔORF24) for their kind gift of viruses and cells. Cell culture models were periodically subjected to mycoplasma testing by PCR or enzymatic reporter assays and found negative.

### Virus stock preparation and titration

HSV-1: Vero-based cell lines were infected at a low MOI (~0.01 PFU/cell) and harvested when cells were sloughing from the sides of the vessel. Supernatant and cell fractions were collected and centrifuged at 4000 × g 4 °C for 10 min. The subsequent supernatant fraction was reserved. The pellet fraction was frozen (−80 °C 20 min)/thawed (37 °C 5 min) for three cycles, sonicated for 1 min (70% power, water bath), and centrifuged at 2000 × g 4 °C for 10 min. The final virus stock was composed of the cell-associated virus and reserved supernatant virus fractions. KOS, strain 17, and n199 were prepared and titered in Vero cells. Other virus stocks were prepared and titered in the following Vero-based complementing cell lines: E5 (ICP4 + , n12-complementing) and E11 (ICP4/ICP27 + , 5dl1.2-complementing).

KSHV: iSLK-BAC16 cells were induced with 1 μg/mL doxycycline and 1 mM sodium butyrate for 3 days. Cell debris was removed from the supernatant fraction by centrifuging at 2000 × g 4 °C for 10 min and filtering with a 0.45 PES membrane. The virus was concentrated after a 16,000 × g 4 °C 24 h spin and resuspended in a low volume of EGM2 media (approx. 1000-fold concentration). To assess viral infectivity, VEC or LEC cells were infected with serial dilutions of the BAC16 stock and assessed using Flow Cytometry for GFP+ cells at 3 days post infection. BAC16 contains a constitutively expressed GFP gene within the viral genome. Based on these assays, BAC16 stock was used at a 1:60 dilution, resulting in 35% infection for HUVEC (MOI 0.25) and 70% infection for LEC (MOI 1).

MHV68: NIH 3T12-based cell lines were infected at a low MOI with H2B-YFP until 50% cytopathic effect was observed. Infected cells and conditioned media were dounce homogenized and clarified at 600 × g 4 °C for 10 min. The clarified supernatant was further centrifuged at 3000 × g 4 °C for 15 min and then 10,000 × g 4 °C for 2 h to concentrate ~40-fold in DMEM. H2B-YFP was prepared and titered using plaque assays in NIH 3T12 cells. The recombinant MHV68 ORF50.stop virus was produced and titered using plaque assays in CS-RTA4 3T12 cells (Li et al, 2017a).

### De novo infection

HSV-1: Confluent MRC-5 cells were infected with ten plaque-forming units (PFU) per cell. Virus was adsorbed in PBS for 1 h at room temperature. Viral inoculum was removed, and cells were washed quickly with PBS before adding on DMEM media supplemented with 2% FBS. 0 h time point was considered after adsorption of infected monolayers when cells were placed at 37 °C to incubate. If indicated, cells were treated with 75 μg/mL Cycloheximide (Sigma #C4859) or 300 μg/mL Phosphonacetic Acid (Sigma #284270) at 0 h post infection.

KSHV: VEC or LEC cells were infected with BAC16 at an approximate MOI of 0.25 (30% cells infected) or 1 (70% cells infected), respectively. Virus was adsorbed in a low volume of media for 8 h at 37 °C, after which viral inoculum was removed and replaced with fresh media. In all, 0 h time point was when virus was added and cells were first placed at 37 °C to incubate.

MHV68: Subconfluent NIH 3T3 fibroblasts were infected with 5 PFU per cell. The virus was adsorbed in a low volume of DMEM media supplemented with 8% FBS for 1 h at 37 °C, prior to overlay

with fresh media. In all, 0 h time point was when virus was first added, and cells were placed at 37 °C to incubate. If indicated, cells were treated with 100 μg/mL phosphonacetic acid at 1.5 h post infection.

### HSV-1 mouse infections

Female 8-week-old BALB/cAnNTac mice were infected with $10^5$ PFU HSV-1 (strain 17) via the ocular route. After 4–5 weeks, latently infected trigeminal ganglia were harvested and immediately processed or explanted into the culture to induce viral reactivation. Trigeminal ganglia explants were cultured (DMEM/1% FBS) for 12 h at 37 °C/5% CO$_2$ in the presence of vehicle (DMSO), 100 μM acyclovir, or 2 μM JQ1 (Cayman Chemical CAS: 1268524-70-4). Pools of six ganglia were homogenized in 1 ml TriPure isolation reagent (Roche) using lysing matrix D on a FastPrep24 instrument (3 cycles of 40 s at 6 m/s). In total, 0.2 ml chloroform was added for phase separation using phase lock gel heavy tubes and RNA isolation from the aqueous phase was obtained by using ISOLATE II RNA Mini Kit (Bioline). RNA quality was verified with Agilent 2100 Bioanalyzer System using RNA Nano Chips (Agilent Technologies). All animal care and handling were done in accordance with the U.S. National Institutes of Health Animal Care and Use Guidelines and as approved by the National Institute of Allergy and Infectious Diseases Animal Care and Use Committee (Protocol LVD40E, TMK).

### MHV68 mouse infections

Mixed bone marrow chimeric mice generated by reconstitution of CD45.1 C57/BL6 recipient mice with bone marrow from CD45.2 *CD19*[cre/+] STAT3 wild-type mice and *CD19*[cre/+]*Stat3*[f/f]*tdTomato*[stopf/f] B-cell STAT3 knockout mice were infected by intraperitoneal injection with 1000 PFU MHV68 H2B-YFP in 0.5 ml under isoflurane anesthesia. At 16 days post infection (dpi), mouse spleens were homogenized, treated to remove red blood cells, and enriched for B cells (Pan B-cell isolation kit; STEMCELL, Vancouver, BC, Canada). YFP+ infected and YFP- uninfected GL7 + CD95+ germinal center (GC) B cells were sorted and collected from tdTomato- (*CD19*[cre/+] wild-type*)* CD3-B220 + CD45.2 + B cells. In total, $5 \times 10^4$ cells sorted from pooled mouse splenocytes were spun down, resuspended in 50 μl of TRIzol, and stored at −80 °C. GENEWIZ (South Plainfield, NJ) performed RNA extraction, quality control, library preparation, and Illumina sequencing. Additional experimental details are included in (Hogan et al, 2024). All mouse experiments were performed in accordance with protocols approved by the National Cancer Institute Animal Care and Use Committee.

### Lytic reactivation

KSHV: Subconfluent monolayers of iSLK-BAC16 (WT, ΔORF24, ΔORF57) were induced with 1 μg/mL doxycycline, 1 mM sodium butyrate in DMEM media supplemented with 2% Tet-approved FBS. 0 h time point was when induction media was added and cells were first placed at 37 °C to incubate. If indicated, cells were treated with 100 μM Cidofovir (Sigma) or 70 μg/mL phosphonacetic acid (Sigma) at 0 h post induction.

MHV68: One day prior to induction, A20 HE-RIT cells were subcultured at a 1:3 dilution in media lacking antibiotics. Cells were seeded subconfluently and induced for 24 h with RPMI media containing 5 μg/ml Dox and 20 ng/ml TPA.

### RNase R confirmation of viral circRNAs

HSV-1: The total RNA was isolated from cells using the Direct-zol RNA MiniPrep Kit (Zymo), following the manufacturer's instructions. 7 μg total RNA was combined with 1 × E-PAP Buffer, 2.5 mM MnCl2, 1 mM ATP, 40 Units RNase Inhibitor (ThermoFisher), and 8 Units E-PAP (PolyA Tailing Kit, ThermoFisher). PolyA tailing reactions were incubated at 37 °C for 10 min. RNA was acid-phenol chloroform extracted, ethanol precipitated, and resuspended in nuclease-free water. PolyA-Tailed material was combined with 20 Units RNase Inhibitor (ThermoFisher), 20 Units RNase R (Lucigen), 100 mM LiCl, 20 mM Tris-HCl pH 8.0, 0.1 mM MgCl$_2$. RNA was RNase R digested at 37 °C for 30 min, following cleanup with the RNA Clean and Concentration Kit (Zymo). RNA was reverse transcribed with random decamers using ReverTra Ace-α qPCR RT Master Mix (Toyobo). cDNA was measured using quantitative PCR (qPCR) with either divergent (circRNA) or convergent (mRNA) primers.

KSHV and MHV68: The total RNA was isolated from cells using the Direct-zol RNA MiniPrep Kit (Zymo), following the manufacturer's instructions. Overall, 3 μg total RNA was combined with 20 Units RNase Inhibitor (ThermoFisher), 20 Units RNase R (Lucigen), 100 mM KCl, 20 mM Tris-HCl pH 8.0, 0.1 mM MgCl$_2$. RNA was RNase R digested at 37 °C for 30 min, following cleanup with the RNA Clean and Concentration Kit (Zymo). RNA was reverse transcribed with random decamers using ReverTra Ace-α qPCR RT Master Mix (Toyobo). cDNA was measured using quantitative PCR (qPCR) with either divergent (circRNA) or convergent (mRNA) primers.

### Splicing inhibitor assays

HSV-1: Confluent monolayers of MRC-5 were infected with HSV-1 (strain KOS) at an MOI of 10 PFU/cell. At 1 or 3 h post infection, the supernatant was removed and replaced with media containing 30 μM Isoginkgetin (CAS #548-19-6, Sigma #416154), 25 nM Pladienolide B (CAS #445493-23-2, Santa Cruz #sc-391691), or vehicle (0.1% DMSO). Total RNA was collected at 24 h post infection.

KSHV: Subconfluent monolayers of iSLK-BAC16 were treated with 1 mM NaB 1 μg/mL Dox to induce lytic reactivation. At 24 h post reactivation, 30 μM Isoginkgetin, 25 nM Pladienolide B, or vehicle (0.1% DMSO) was added to the media. Total RNA was collected at 48 h post reactivation.

### Loss-of-function studies

HSV-1: Subconfluent dishes of MRC-5 were incubated in OptiMEM (ThermoFisher #31985062) with 50 nM siRNA and Lipofectamine RNAiMAX Transfection Reagent (ThermoFisher #13778150). For dual knockdown, the total siRNA concentration was 50 nM which was composed of 25 nM RLIG1 siRNA + 25 nM RTCB siRNA. siRNAs were ON-TARGETplus SMARTpools for Human RLIG1 (Horizon #L-032720), RTCB (Horizon #L-017647), DBR1 (Horizon #L-008290), or Non-targeting control (Horizon #D-001810). After incubation at 37 °C for 8 h, transfection media was removed and replaced with 1 × DMEM 2% FBS. At 2 days post transfection, MRC-5 were infected with HSV-1 strain KOS at an MOI of 10 PFU/cell and RNA collected at either 12 or 24 h post infection.

KSHV: Subconfluent dishes of iSLK-BAC16 were incubated in OptiMEM (ThermoFisher #31985062) with 50 nM siRNA and

Lipofectamine RNAiMAX Transfection Reagent (ThermoFisher #13778150). For dual knockdown, total siRNA concentration was 50 nM which was composed of 25 nM RLIG1 siRNA + 25 nM RTCB siRNA. siRNAs were ON-TARGETplus SMARTpools for Human RLIG1 (Horizon #L-032720), RTCB (Horizon #L-017647), DBR1 (Horizon #L-008290), PRPF8 (Horizon #L-012252), or Non-targeting control (Horizon #D-001810). For PRPF8 loss-of-function experiments, iSLK-BAC16 were treated with 1 mM sodium butyrate 1 μg/mL doxycycline at 8 h post transfection and RNA collected at 72 h post transfection. For all other knockdown targets, iSLK-BAC16 were reactivated at 24 h post transfection and RNA collected at 96 h post transfection.

### Relative (qPCR) quantification of transcripts

The total RNA was isolated from cells using the Direct-zol RNA MiniPrep Kit (Zymo #R2053), following the manufacturer's instructions. Unless otherwise specified, total RNA was reverse transcribed with random decamers using the ReverTra Ace-α qPCR RT Master Mix (Toyobo #FSQ-201). For the detection of lariat species, RNA was reverse transcribed with random decamers and 0.5 μM branchpoint primers (ID1_Lariat_BP, DKK1_lariat_BP) using the ReverTra Ace-α qPCR RT Master Mix. For detection of spliced targets—mature-ORF57, mature-ICP0, mature-UL15, mature-BRD2, mature-DNAJB1—RNA was reverse transcribed with polydT oligos (Invitrogen #N8080128) using the High-Capacity cDNA Reverse Transcription Kit (Applied Biosystems #4374966). qPCR was performed using either THUNDERBIRD Probe qPCR Master Mix (DiagnoCine #TYB-QPS-101) or THUN-DERBIRD Next Syber qPCR Master Mix (DiagnoCine #TYB-QPX-201).

### Absolute (ddPCR) quantification of transcripts

The total RNA was isolated from cells using the Direct-zol RNA MiniPrep Kit (Zymo #R2053), following the manufacturer's instructions. In all, 0.5–2 μg of total RNA was reverse transcribed with random decamers using the ReverTra Ace-α qPCR RT Master Mix (Toyobo #FSQ-201). cDNA was measured with either divergent (circRNA) or convergent (mRNA) primers using the QX200 Droplet Digital PCR System (Bio-Rad).

### Quantification of viral genomes

The cell fraction was isolated from infection models. Cell pellets were washed with 1 × PBS and lysed using 0.5% SDS, 400 μg/mL proteinase K, 100 mM NaCl. Samples were incubated at 37 °C for 12–18 h and heat-inactivated for 30 min at 65 °C. DNA samples were serial diluted 1:1000 and measured using qPCR with primers specific to HSV-1 UL23, KSHV ORF6, MHV68 ORF50, human GAPDH, or mouse GAPDH. Standard curves were generated using purified genomic stocks (HSV-1 BAC, KSHV BAC, human genome Promega #G1471, mouse genome Promega #G3091), or purified plasmids (MHV68 ORF50). The absolute copy number of genomic stocks was determined using ddPCR. Values were plotted as follows: $viral\ genomes/cell = \frac{viral\ gene\ copy\ number}{host\ gene\ copy\ number/2}$.

### Quantification of infectious HSV-1 progeny

The supernatant fraction was collected and freeze-thawed three times. Vero cells were seeded at a density of $10^6$ cells/well in a six-well dish and infected with a serially diluted supernatant. Viral yield was determined by plaque assay and plotted as the total PFU for the entire volume of supernatant collected.

### Agarose gel assessment of host splicing

We adapted a previously published method, for detection of DNAJB1 transcripts (Chang et al, 2021). The total RNA was reverse transcribed using the ReverTra Ace-α qPCR RT Master Mix and random decamers. In total, 20 ng cDNA was PCR amplified using 2× KOD One PCR Master Mix (Toyobo # KMM-101) and amplified with the following cycling conditions: 94 °C 5 min, (94 °C 30 s, 60 °C 30 s, 68 °C 2 min) × 35 cycles, 72 °C 5 min. PCR material was purified using the Qiagen MinElute kit, run on a 2% agarose 1 × TBE gel, and visualized with ethidium bromide staining.

### Cell viability assays

MRC-5 or iSLK-BAC16 cells were treated with 10, 30, 60 μM Isoginkgetin (CAS #548-19-6, Sigma #416154) or 12.5, 25, 50 nM Pladienolide B (CAS #445493-23-2, Santa Cruz #sc-391691). When indicated (vehicle), DMSO (0.1%) was added to media. After 24 h of drug treatment, CellTiter-Glo 2.0 Reagent (Promega #G9241) was added 1:1 to media. Luminescence was measured using the Modulus II Multiplate reader (Turner Biosystems). All values are the average of at least three biological replicates, and values were calculated relative to the luminescence of the unpaired "vehicle" sample.

### Immunoblotting

Cell monolayers were washed twice with 1 × PBS 0.1 mM Tosyllysine Chloromethyl Ketone (TLCK) and then scraped into 1 mL 1 × PBS 0.1 mM TLCK. Cell suspensions were transferred to a microfuge tube and pelleted at 2000 × g for 5 min at 4 °C. Cell pellets were resuspended in 1 × RIPA (Sigma-Aldrich #R0278) containing 1 × protease inhibitors (Roche #11836170001). Cells were mixed by gently pipetting followed by a 15 min incubation on ice. Samples were centrifuged at 16,000 × g 10 min 4 °C, and the supernatant fraction was retained. Protein lysates were assessed via immunoblot using the XCell SureLock Mini-Cell and XCell II Blot Module (ThermoFisher #EI0002). Antibodies used in this study include VCL (1:5000 dilution), DBR1 (1:200 dilution), RTCB (1:500 dilution), RLIG1 (1:250 dilution), ICP8 (1:1000 dilution), ORF57 (1:200 dilution), ORF6 (GeneTex custom synthesis antibody, 1:200 dilution), GAPDH (1:3000 dilution), LI-COR IRDyes (1:2000 dilution). Band intensities were quantified in Image Studio (LI-COR, version 5.2.5), relative to a reference protein (GAPDH or VCL) in the same sample.

### Amplicon confirmation

Total RNA was isolated from cells using the Direct-zol RNA MiniPrep Kit (Zymo), following the manufacturer's instructions. 7 μg total RNA was combined with 1 × E-PAP Buffer, 2.5 mM MnCl$_2$, 1 mM ATP, 40 Units RNase Inhibitor (ThermoFisher), and 8 Units E-PAP (PolyA Tailing Kit, ThermoFisher). PolyA tailing reactions were incubated at 37 °C for 10 min.

RNA was purified with the RNA Clean and Concentration Kit (Zymo), and resuspended in nuclease-free water. PolyA-Tailed material was combined with 20 Units RNase Inhibitor (Thermo-Fisher), 20 Units RNase R (Lucigen), 100 mM LiCl, 20 mM Tris-HCl

pH 8.0, 0.1 mM MgCl$_2$. RNA was RNase R digested at 37 °C for 30 min, following cleanup with the RNA Clean and Concentration Kit (Zymo). RNA was reverse transcribed with random decamers using ReverTra Ace-α qPCR RT Master Mix (Toyobo). ~25 ng cDNA was PCR amplified using either KOD One (Toyobo) or FailSafe (BioSearch Technologies) Enzyme mixes. PCR products were purified using Monarch PCR & DNA Cleanup Kit (NEB). Amplicons were either run on an TapeStation using a D1000 screentape (Agilent) or pooled and sent to Plasmidsaurus Inc. for sequencing.

### Ribominus total RNA-Seq

The total RNA was isolated from cells using the Direct-zol RNA MiniPrep Kit (Zymo), following the manufacturer's instructions. ERCC spike-in controls (ThermoFisher) were added to 500–1000 ng of total RNA. RNA was sent to the NCI CCR-Illumina Sequencing facility for library preparation and sequencing. RNA was ribominus selected and directional cDNA libraries were generated using either Stranded Total RNA Prep with Ribo-Zero Plus (Illumina # 20040525) or TruSeq Stranded Total RNA Ribo-Zero Gold (Illumina #RS-122-2303). In all, 2–4 biological replicates were sequenced for all samples. Sequencing was performed at the NCI CCR Frederick Sequencing Facility using the Illumina NextSeq 550 or Illumina NovaSeq SP platform to generate 150 bp PE reads.

### 4sU-sequencing

iSLK-BAC16 cells were uninduced or induced with 1 µg/mL Doxycycline 1 mM Sodium Butyrate. At indicated times relative to induction, 10 mM 4sU (Sigma) was added to cell culture medium. 15 min post-4sU addition, cells were collected and RNA extracted using Direct-zol RNA MiniPrep Plus (Zymo). In total, 40–50 µg total RNA was biotinylated in 10 mM Tris pH 7.4, 1 mM EDTA, 0.2 mg/mL EZ-link Biotin-HPDP (ThermoFisher). Unbound biotin was removed by performing a chloroform:isoamyl alcohol extraction using MaXtract High-Density tubes (Qiagen). RNA was isopropanol precipitated and resuspended in water. Biotinylated RNA was bound 1:1 to Dynabeads My One Streptavidin T1 equilibrated in 10 mM Tris pH 7.5, 1 mM EDTA, 2 M NaCl. Bound beads were washed three times with 5 mM Tris pH 7.5, 1 mM EDTA, 1 M NaCl. 4sU RNA was eluted with 100 mM DTT and isolated using the RNeasy MinElute Cleanup Kit (Qiagen). RNA was sent to the NCI CCR-Illumina Sequencing facility for library preparation and sequencing. Pulldown RNA was ribominus selected using the NEBNext rRNA Depletion Kit v2 (NEB # E7400L) and RNA-Seq libraries were generated using the NEBNext Ultra II Directional RNA Library Prep Kit (NEB # E7760L). Two biological replicates were sequenced using the Illumina NextSeq 550 platform to generate 150 bp PE reads.

### eCLIP sequencing

Subconfluent monolayers of iSLK-BAC16 were induced for 24 h with 1 µg/mL Doxycycline 1 mM Sodium Butyrate in DMEM media supplemented with Tet-approved FBS. ORF57 eCLIP was performed by Eclipse Bioinnovations (San Diego, CA) following the published protocol (Van Nostrand et al, 2016). 10% of ORF57 immunoprecipitations, and 1% of inputs were run on NuPAGE 4–12% Bis-Tris protein gels and transferred to a nitrocellulose membrane. Membranes were probed using ORF57 (1:6000 dilution) primary antibody and EasyBlot anti-Rabbit IgG (1:10,000

dilution) and imaged with C300 Imager using Azure Radiance ECL. ORF57 was detected by western blot running around 75 kDa, so IP and size-matched input were taken from 51 kDa to 126 kDa regions. For these assays, two independent biological replicates were performed using an affinity-purified anti-ORF57 polyclonal antibody (Sei and Conrad, 2011; Sei et al, 2015). An IgG control antibody was tested in parallel, but the yield from this was low, so all subsequent comparisons were performed relative to the size-matched input control.

## Bioinformatic analysis

### De novo circRNA annotation

Illumina SRS data was analyzed using Circrnas in Host And viRuses anaLysis pIpEline (CHARLIE) v.0.9.0, source code available at https://github.com/CCBR/CHARLIE. The workflow is brief as follows. RNA-sequencing reads were trimmed using CutAdapt (Martin, 2011). Trimmed reads were mapped using STAR (2-pass mapping) to concatenated genome assemblies which contain the host (hg38 or mm39) + virus (KT899744.1, NC_009333.1, MH636806.1) + ERCC spike-in controls (Dobin et al, 2012). Our pipeline combines five previously published tools for circRNA discovery, CIRCexplorer2 (Zhang et al, 2016), CIRI2 (Gao et al, 2017), DCC (Cheng et al, 2015), circRNAFinder (Westholm et al, 2014), and find_circ (Memczak et al, 2013). BSJ calls must meet the following criteria: ≥3 counts, 18 nucleotides (nt) mapped on both sides of the BSJ, circRNA splice donor–acceptor distance ≥150 nt, viral circRNA splice donor–acceptor distance ≤5000 nt. Viral circRNA splice donor–acceptor distance was limited to 5 kb to prevent artefactual calls caused by the isomeric structure of the HSV-1 genome. For PE data, BSJ were further filtered, requiring the mate be contained within the 5' and 3' junction of the chimeric mapped read. High-confidence circRNAs were those in agreement between CIRCexplorer2 STAR (Dobin et al, 2012) and BWA (Li and Durbin, 2009) mapping outputs and called in at least one other tool. These criteria were in line with recent recommendations (Chen et al, 2020; Hansen, 2018) for best circRNA practices, requiring agreement between at least two de novo circRNA annotation tools reliant on distinct RNA-Seq mapping programs. Importantly, this approach enables de novo circRNA calling for genome assemblies lacking curated transcript features, e.g., transcription start sites, transcription end sites, exons, introns, as is the case for viral genomes.

CircRNAs are labeled based on their colinear genes, chromosome, and 5' and 3' BSJ positions. Genomic BSJ positions were used to determine circRNA species overlapping between conditions. To visualize data, STAR-mapped BAMs were filtered for "linear" and "circular" reads. "Linear" BAMs included all reads, except those spanning BSJ. "circular" BAMs included reads spanning a BSJ. Bigwigs were generated using deepTools2 bamcoverage and bigwigCompare with 1 bp bins (Ramírez et al, 2016). Transcript traces were visualized using Integrative Genomics Viewer (IGV) (Robinson et al, 2011).

### Transcriptome analysis

Standard gene expression analysis: RNA-Sequencing reads were trimmed using Cutadapt (Martin, 2011) and the following parameters: --pair-filter=any, --nextseq-trim=2, --trim-n, -n 5, --max-n 0.5, -0 5, -q 20, -m 15. Trimmed reads were mapped using

STAR (Dobin et al, 2012) with 2-pass mapping to concatenated genome assemblies which contain the host genome (hg38 or mm39) + virus genome (KT899744.1, NC_009333.1, MH636806.1) + ERCC spike-in controls (Dobin et al, 2012). Details on mapping assemblies are included below. RNA STAR mapping parameters are as follows: --outSJfilterOverhangMin 15 15 15 15, --outFilterType BySJout, --outFilterMultimapNmax 20, --outFilterScoreMin 1, --outFilterMatchNmin 1, --outFilterMismatchNmax 2, --outFilterMismatchNoverLmax 0.3, --outFilterIntronMotifs None, --alignIntronMin 20, --alignIntronMax 2000000, --alignMatesGapMax 2000000, --alignTranscriptsPerReadNmax 20000, --alignSJoverhangMin 15, --alignSJDBoverhangMin 15, --alignEndsProtrude 10 ConcordantPair, --chimSegmentMin 15, --chimScoreMin 15, --chimScoreJunctionNonGTAG 0 --chimJunctionOverhangMin 18, --chimMultimapNmax 10. STAR GeneCount (per gene read counts) files were used for transcript quantitation. 4sU-Seq data was normalized as reads per million total reads per kilobase pair (RPKM). For all other data, ERCC reads were used to generate standard curves similar to (Schertzer et al, 2020), using their known relative concentrations. All biological replicates had ERCC-derived standard curves with $R^2 > 0.9$. ERCC normalized gene counts were calculated as follows:

$$\textbf{Log}_2\textbf{RPKM} : Log_2\left(\frac{Raw\ gene\ Counts}{gene\ size\ in\ kb \times Million\ total\ reads}\right)$$

*Raw gene counts include forward spliced reads and exclude reads containing BSJ.

$$\textbf{ERCC norm.gene counts} : \left(2^\wedge\left(\frac{Log_2(RPKM) - ERCC\ derived\ intercept\ (b)}{ERCC\ derived\ slope\ (y)}\right)\right)/10,000$$

Viral transcript kinetic classes: HSV-1 transcripts were clustered into the following temporal classes: Immediate Early (IE)-UL54, US1, US12, RL2, RS1; Early (E)-UL2, UL12, UL13, UL23, UL29-30, UL39-40, UL50; Leaky Late (L1)-UL4-9, UL11, UL14-15, UL17-22, UL24, UL26-28, UL32-35, UL41, UL42-43, UL45-46, UL48-49, UL52-53, UL55-56, US3-4, US6-9; True Late (L2)-UL1, UL3, UL10, UL16, UL25, UL31, UL36-38, UL44, UL47, UL49A, UL51, US2, US5 (Dremel and DeLuca, 2019). KSHV transcripts were clustered into the following temporal classes: Immediate Early (IE)-ORF50, Hygromycin B and GFP (EF1α-promoter cassette); Early (E)-ORF2, ORF4, ORF6-7, ORF10-11, ORF57-61, ORF63, ORF68-72, LANA, PAN, K1–4, K4.1, K5-6, K8, K12; Leaky Late (L1)-ORF9, ORF16-18, ORF20-24, ORF28-32, ORF34-38, ORF40, ORF42-44, ORF46-49, ORF54-56, ORF62, ORF64, ORF66-67, ORF67A, ORF74-75, K4.2, K7, K14, K15, vIRF1-4; Late (L)-ORF8, ORF19, ORF25-27, ORF33, ORF39, ORF45, ORF52-53, ORF65, K8.1 (Arias et al, 2014; Dremel and Didychuk, 2023). MHV68 transcripts were clustered into the following temporal classes: Immediate Early (IE)-ORF50; Early (E)-M4, ORF6, ORF10, ORF30, ORF36-37, ORF44, ORF48, ORF54, ORF56-57, ORF59-61, ORF75A; Leaky Late (L1)-M1, M3, M5, M6, M14, ORF7, ORF9, ORF11, ORF18, ORF21, ORF24, ORF27, ORF32, ORF34-35, ORF38, ORF40, ORF42, ORF46, ORF58, ORF63, ORF68, ORF72; Late (L)-M7, M9, M15, M17, M18, ORF4, ORF8, ORF17, ORF19-20, ORF22-23, ORF26, ORF29, ORF33, ORF39, ORF43, ORF45, ORF49, ORF52-53, ORF55, ORF62B, ORF64, ORF66-67, ORF6–9 (Cheng Benson Yee et al, 2012; O'Grady et al, 2019).

CircRNA quantitation: CHARLIE was used to map, annotate, and filter high-confidence BSJ as described above. BSJ were quantified from STAR output bams and normalized as RPKM (4sU-Seq) or relative to ERCC spike-in controls (all other data). Gene length for circRNA was treated as 0.15 kb as that is the total read length and full circRNA size is unknown.

Circ/linear transcript quantitation: High-confidence BSJ identified by CHARLIE were quantified from STAR output bams for circRNA and colinear transcripts. CircRNA reads were those that spanned BSJ. Linear reads were those which spanned FSJ in addition to all non-chimeric reads. Circ to linear ratios are the raw reads for circ over linear.

### Amplicon-sequencing analysis

Sequencing was performed by Plasmidsaurus Inc. using Oxford Nanopore Technologies long-read sequencing (v14 library prep chemistry, R10.4.1 flow cell). Raw data was delivered in fastq.gz format. Sequences were filtered by size, only continuing with those between 50 and 500 bp for downstream analysis. Amplicon primers were used to demultiplex pooled samples. If needed, 3' A's were trimmed as these were byproducts of the library synthesis. In some cases 5' barcodes were appended to forward primers, these were trimmed prior to subsequent analysis. Antisense amplicons were reverse complemented and merged with those derived from the sense strand. Unless indicated, all amplicons containing the forward and reverse primers were used to generate a consensus sequence. In select cases (HSV-1 circUL29, circUL42, circUL44, KSHV circPAN, circK2, circK5, circORF61, MHV68 circORF66, circM7, circORF6) amplicon pools were subsampled by size, this was necessary as some primers amplified multiple BSJ junctions. Amplicon consensuses were generated using SnapGene v.7.2.1 MAFFT v.7.471. All amplicon consensus sequences are reported in Dataset EV2. Amplicon consensuses were scored for mismatches and gaps relative to the expected sequence (% match with expected).

### Flanking cis-element analysis

Splice donor–acceptor frequency: The two nucleotides immediately external to the BSJ were determined for all high-confidence circRNAs. For frequency calculations, the splice DA sequences was multiplied by the abundance of a given circRNA in the sample. Thus, circRNA abundance affects total frequency calculations.

GC content: The 100 nucleotides (100 nt 5' + 100 nt 3') immediately external to the BSJ were determined for all high-confidence circRNAs. Flanking sequences were multiplied by the abundance of a given circRNA in the sample. GC content was determined using geecee (Galaxy Version 5.0.0) (Afgan et al, 2018). To calculate viral gene theoretical GC content, we used the CDS coordinates of KT899744.1 (HSV-1), NC_009333.1 (KSHV), and MH636806.1 (MHV68). To calculate host gene theoretical GC content, we used the gene coordinates of hg38 gencode.v36 and mm39 gencode.vM29 (Frankish et al, 2020).

Motif enrichment and RBP predictions: The 100 nucleotides (100 nt 5' + 100 nt 3') immediately external to the BSJ were determined for all high-confidence circRNAs. Motif discovery and comparison were performed using MEME suite (Bailey et al, 2015). Motif discovery was performed using MEME v5.5.4 with the following parameters: -oc. -nostatus -time 14400 -mod anr -nmotifs 5 -minw 6 -maxw 10 -objfun classic -minsites 2 -revcomp -markov_order 1. Only motifs with $P$ values < 0.05 were reported. Predicted RBP partners were determined using Tomtom v5.5.4 (Gupta et al, 2007) with the following parameters: -no-ssc -oc. -verbosity 1 -min-overlap 5 -mi 1 -dist pearson -evalue -thresh 10.0 -time 300. Motifs were searched using their species-

appropriate CISBP-RNA homo sapien or mus musculus database (Ray et al, 2013). Only RBP with p values < 0.05 were reported.

### In silico prediction of circRNA–miRNA-mRNA networks

Expected full-length sequences (sequence between 3' and 5' back splice junctions) for viral latency circRNAs were used as input. Prediction of miRNA binding sites was performed using RNAHybrid, with a Gibbs free energy threshold (circLAT: -30, circK12: -10, circORF66: -25), disallowing G:U in seed, and helix constraint from 2 to 8 (Rehmsmeier et al, 2004). miRNA-mRNA interactions were determined using QIAGEN Ingenuity Pathway Analysis (IPA) v.01-22-01 MicroRNA Target Filter. We filtered mRNA targets by those with experimentally observed or high predicted values. mRNA networks were generated using IPA (Krämer et al, 2014) expression analysis. circRNA–miRNA-mRNA network diagrams were made in Cytoscape (Shannon et al, 2003).

### Genome assemblies

HSV-1: KT899744.1, coding sequence (CDS) annotation used for transcript quantification.

KSHV: NC_009333.1, CDS annotation used for transcript quantification.

MHV68: MH636806.1 (O'Grady et al, 2019) modified to remove the beta-lactamase gene (Δ103,908-105,091), CDS annotation used for transcript quantification.

Human: hg38, gencode.v36 (Frankish et al, 2020).

Mouse: mm39, gencode.vM29 (Frankish et al, 2020).

ERCC Spike-In: available from ThermoFisher (#4456740).

## Data availability

HSV-1 infection RNA-Seq data (SRP383035): Sequence Read Archive (SRA) database SRR19779319, SRR19779318, SRR19779317, SRR19779316, SRR19779315, SRR19779314, SRR19779313, SRR19779311. HSV-1 infection with spliceosome inhibition RNA-Seq data (SRP383161): SRA database SRR19787559, SRR19787558, SRR19787557, SRR19787556, SRR19787555. HSV-1 infection with RNA ligase depletion RNA-Seq data (SRP552052): SRA database SRR31749889, SRR31749888, SRR31749887. HSV-1 murine infection RNA-Seq data (SRP383241): SRA database SRR19792335, SRR19792334, SRR25824398, SRR25824397, SRR25824395, SRR25824394, SRR25824396. KSHV infection RNA-Seq data (SRP385335): SRA database SRR20020770, SRR20020769, SRR20020764, SRR20020763, SRR20020762, SRR25816556, SRR20020765, SRR20020766, SRR20020767, SRR20020768, SRR20020757, SRR20020758, SRR20020759, SRR20020760, SRR20020761. KSHV lytic reactivation with spliceosome inhibition RNA-Seq data (SRP383271): SRA database SRR19793315, SRR19793317, SRR19793314. KSHV lytic reactivation with RNA ligase depletion RNA-Seq data (SRP552055): SRA database SRR31749941, SRR31749940, SRR31749939. KSHV lytic reactivation 4sU-pulldown RNA-Seq data (SRP383242): SRA database SRR19792341, SRR19792340, SRR19792338, SRR19792337, SRR19792339, SRR19792336. ORF57 eCLIP sequencing data (SRP383269): SRA database SRR19793302, SRR19793303. MHV68 infection RNA-Seq data (SRP383239): SRA database SRR19792326, SRR19792325, SRR19792322, SRR19792323, SRR19792324, SRR19792321. MHV68 murine infection RNA-Seq data: Gene Expression Omnibus (GEO) database GSE227764.

The source data of this paper are collected in the following database record: biostudies:S-SCDT-10_1038-S44318-025-00398-0.

## Peer review information

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

## Acknowledgements

This work was supported with funds from the National Institutes of Health, Division of Intramural Research NIAID, TMK, ZIA AI000712; NCI, LTK, ZIA BC011953; NCI, JMZ, ZIA BC011176. Additional grant support R01-AI123165 to NKC. The funders had no role in study design, data collection and analysis, decision to publish, or preparation of the manuscript. The authors would like to thank Neal DeLuca (University of Pittsburgh) and Takanobu Tagawa (University of Edinburgh) for thoughtful discussions related to this article. Additional thanks to the National Cancer Institute CCR Genomics Core and CCR-Illumina Sequencing facility for technical support. The resources of the NIH High-Performance Computing Biowulf Cluster were utilized for all computational needs.

## Author contributions

**Sarah E Dremel**: Conceptualization; Data curation; Software; Formal analysis; Validation; Investigation; Visualization; Methodology; Writing—original draft; Project administration; Writing—review and editing. **Vishal N Koparde**: Data curation; Software; Validation; Investigation; Methodology; Writing—review and editing. **Jesse H Arbuckle**: Resources; Investigation; Methodology; Writing—review and editing. **Chad H Hogan**: Resources; Investigation. **Thomas M Kristie**: Resources; Supervision; Funding acquisition; Methodology; Writing—review and editing. **Laurie T Krug**: Resources; Supervision; Funding acquisition; Methodology; Writing—review and editing. **Nicholas K Conrad**: Resources; Data curation; Supervision; Funding acquisition; Investigation; Methodology; Writing—review and editing. **Joseph M Ziegelbauer**: Conceptualization; Supervision; Funding acquisition; Project administration; Writing—review and editing.

Source data underlying figure panels in this paper may have individual authorship assigned. Where available, figure panel/source data authorship is listed in the following database record: biostudies:S-SCDT-10_1038-S44318-025-00398-0.

## Funding

## Disclosure and competing interests statement

The authors declare no competing interests.

# Expanded View Figures

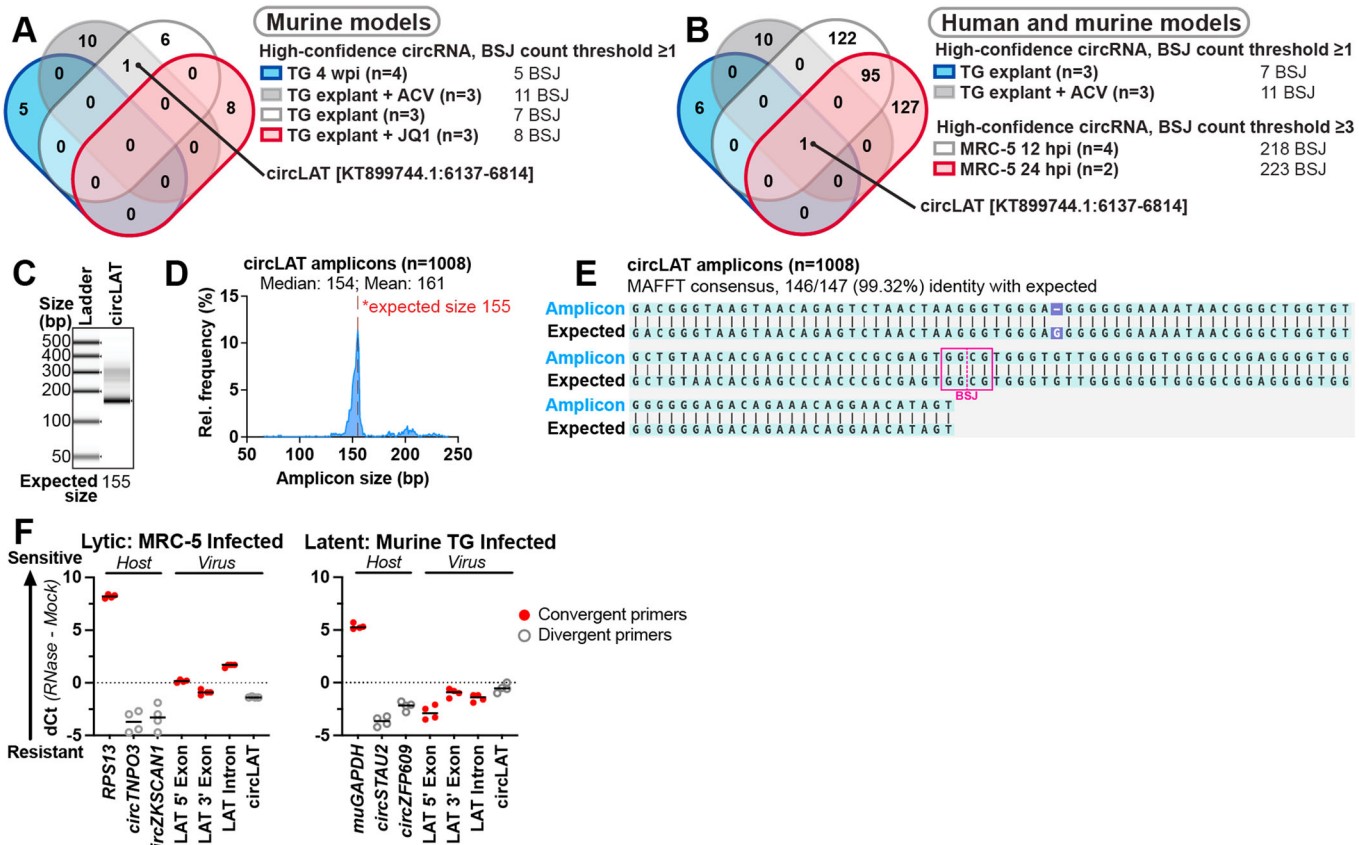

### Figure EV1.  CircRNA derived from HSV-1 latency-associated transcript.

(A, B) Our high-confidence circRNA threshold was lowered (count of ≥1 rather than ≥3) to reanalyze murine HSV-1 infection models. The overlapping incidence of these circRNAs is demonstrated for murine and human models. (C–E) RNase R-digested RNA (MRC-5 12 hpi) was reverse transcribed and PCR amplified with divergent primers. Amplicons were assessed via electrophoresis (TapeStation) or long-read sequencing (Oxford Nanopore Technology). The MAFFT consensus for circLAT was aligned to the expected BSJ sequence identified by CHARLIE, percent matching is reported. (F) RNase R protection assay for RNA from infected MRC-5 (12 hpi) or infected murine TG (4 wpi). cDNA samples were amplified with divergent (gray) or convergent (red) primers. Values are delta Ct (RNase R - Mock Ct), data points are biological replicates ($n = 4$) and horizontal lines are the average.

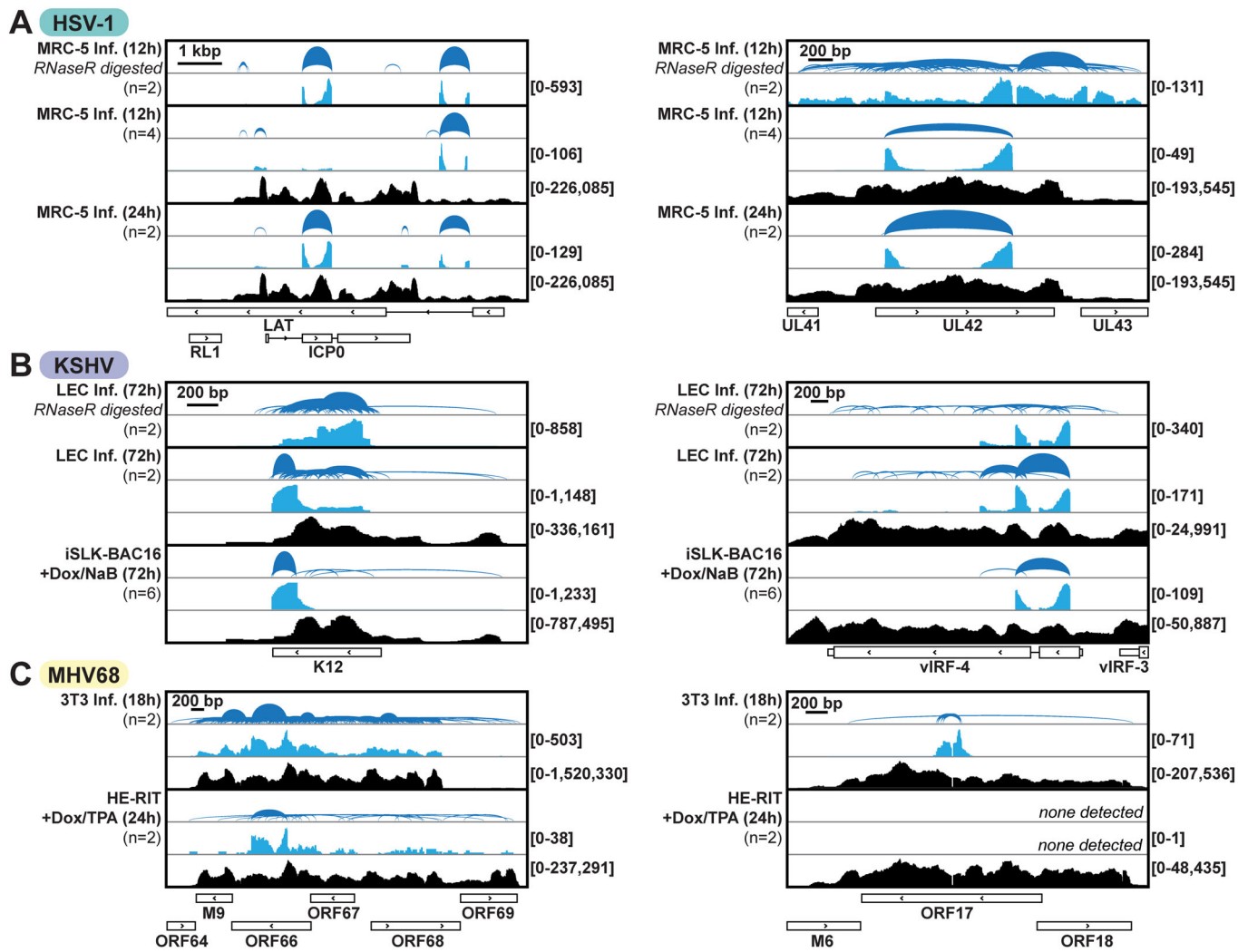

**Figure EV2. Prominent examples of herpesvirus circRNAs.**

(A–C) Visualization of high-confidence circRNAs for HSV-1, KSHV, and MHV68 in lytic models described in Fig. 1. If indicated, RNA samples were treated with RNase R. Viral genes are shown below. Data Information: RNA-Seq was performed and high-confidence circRNAs were called using CHARLIE. Sashimi plots show high-confidence circRNA with arcs proportional to raw BSJ counts. Blue and black traces include circular (back-spliced reads) and linear (non-chimeric) reads, respectively. Traces are the sum of raw BSJ or linear read values for all biological replicates. Y axis minimum and maximum values are shown on the right.

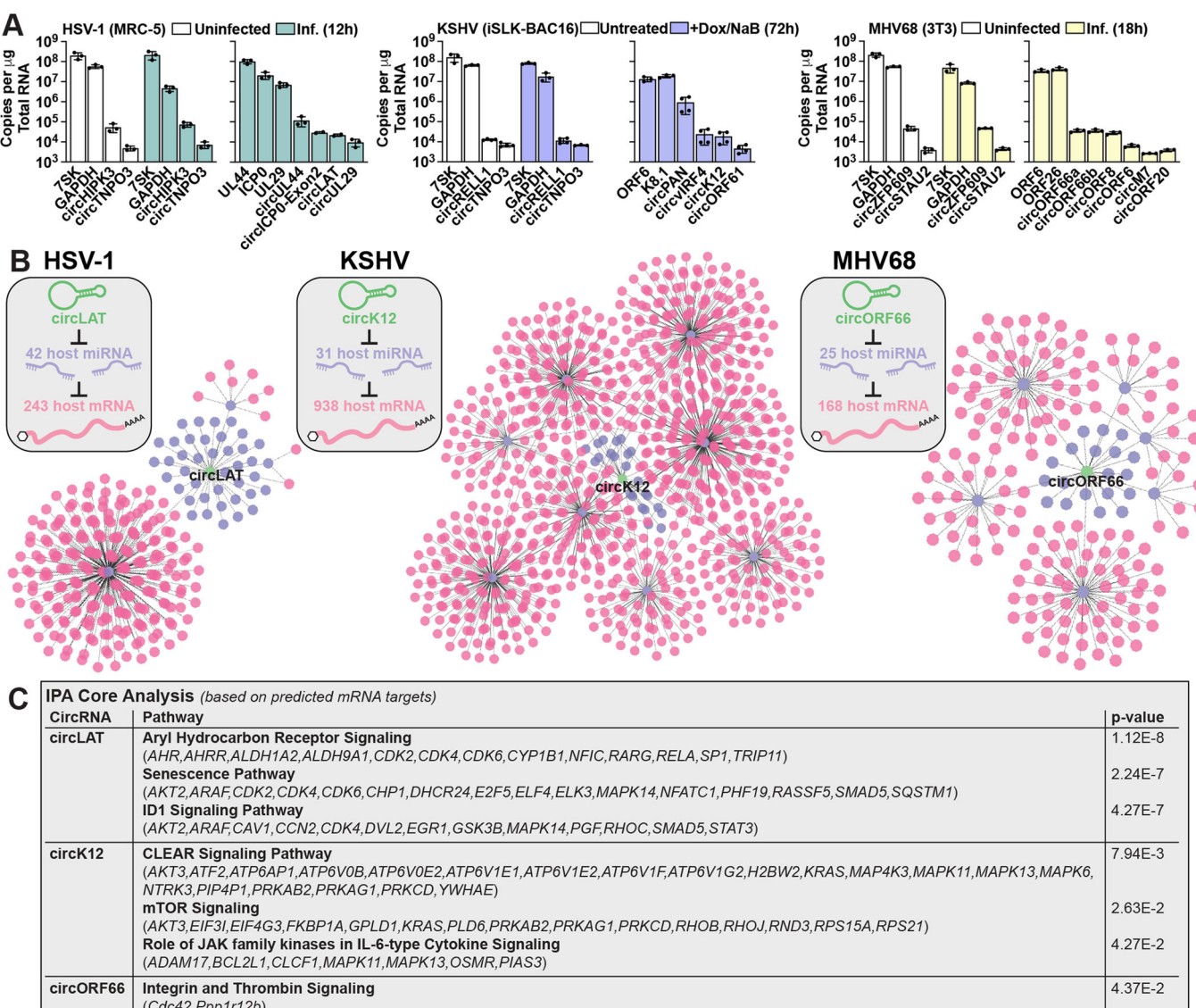

**Figure EV3. Herpesvirus circRNA copy levels and in silico interaction networks.**

(A) Copy level quantitation of viral circRNAs ($n = 3$–4). RNA was collected from the models described in Fig. 1. cDNA was quantified using digital droplet PCR (ddPCR) and convergent (linear transcripts) or divergent (circular transcripts) primers. Data is normalized to the total amount of RNA in the reverse transcription reaction and plotted as copies per µg total RNA. (B) In silico circRNA–miRNA-mRNA interaction networks for circLAT (HSV-1), circK12 (KSHV), and circORF66 (MHV68) variants highlighted in Fig. 1C, D. (C) Putative downstream mRNA targets were used to perform overrepresentation analysis. Pathway hits, mRNA targets present, and p values are given. Data Information: In column bar graphs, data points are biological replicates, bar maxima are the average, error bars are standard deviation. Ingenuity Pathway Analysis (IPA) core analysis $P$ values are calculated using a Fisher's Exact Test.

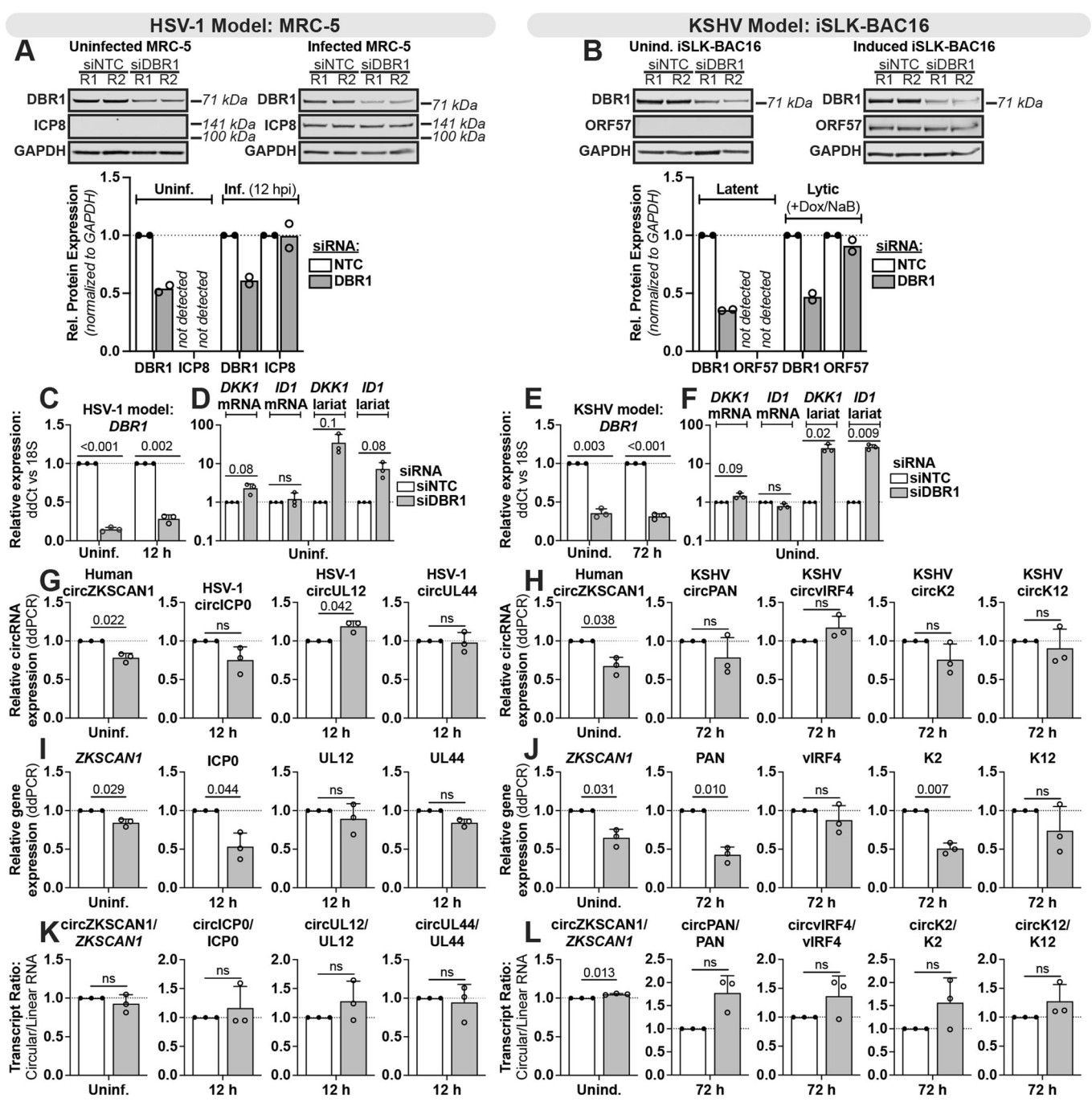

**Figure EV4. Viral circRNAs are refractory to RNA lariat debranching enzyme depletion.**

(A, C, D, G, I, K) MRC-5 were transfected with siRNAs targeting RNA lariat debranching enzyme (*DBR1*) or a nontargeting control (NTC) for 2 days. MRC-5 were mock (Uninf.) or HSV-1 infected for an additional 12 h. (B, E, F, H, J, L) iSLK-BAC16 were transfected with siRNAs targeting *DBR1* or a NTC for 24 h. Subsequently, cells were treated with vehicle (Unind.) or Dox and NaB for 72 h (72 h). (A, B) Protein expression was assessed by immunoblotting and quantified relative to a loading control (GAPDH). (C, F) Host transcripts were quantified by qPCR relative to the reference gene (18S). (G–L). ddPCR quantitation using divergent (circRNA) or convergent (gene) primers. Data Information: In column bar graphs, data points are biological replicates (*n* = 2–3), bar maxima are the average, for *n* > 2 error bars are standard deviation. All data is relative a paired siNTC sample. If *n* ≥ 3, two-tailed paired *t* tests were performed, *P* values < 0.05 are labeled and not significant (ns) indicates *P* values ≥ 0.05. Source data are available online for this figure.

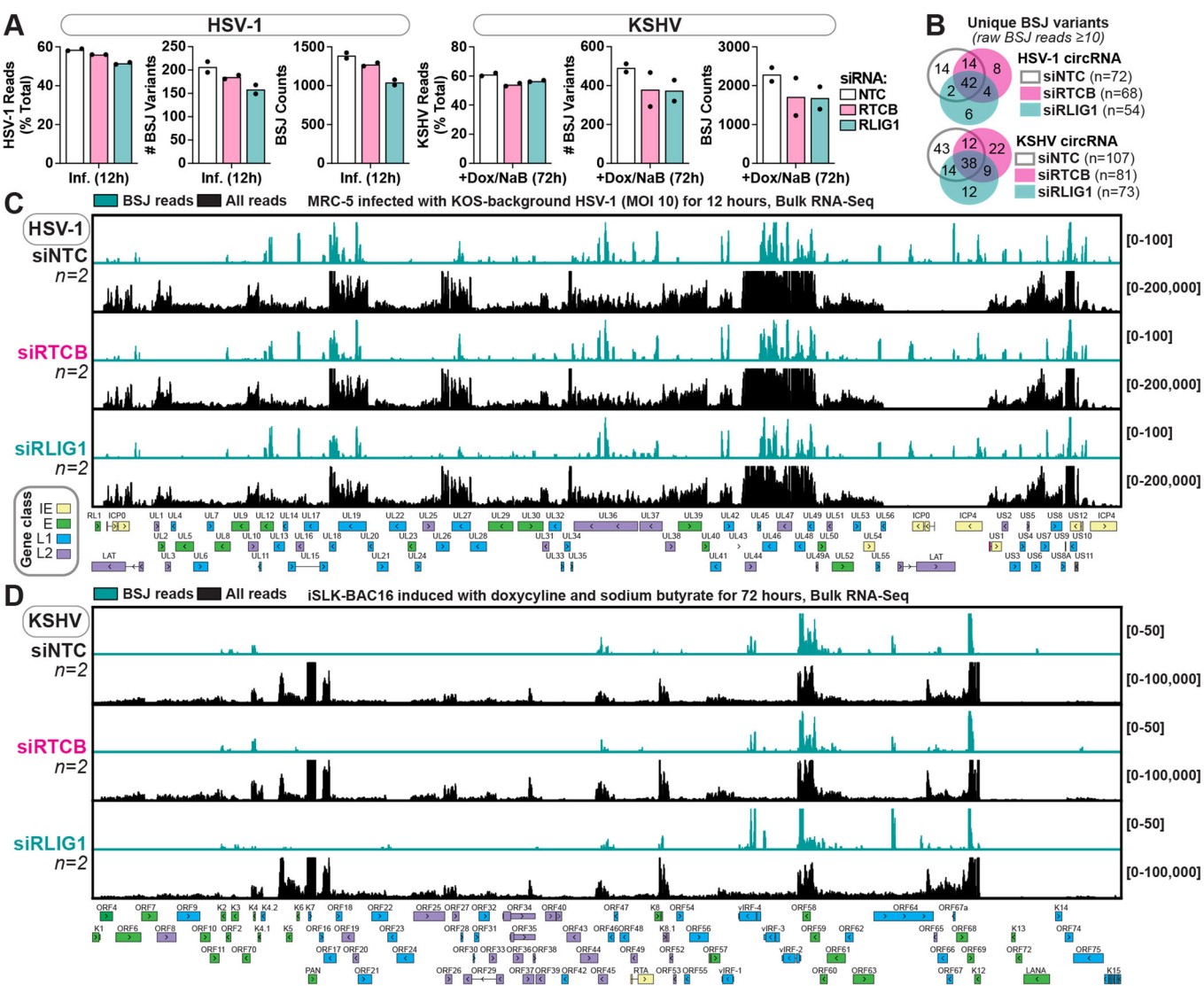

**Figure EV5. Quantitation of viral RNAs after RNA ligase depletion.**

(A–D) RNA-Seq data from infection models in Fig. 5. (A) RNA-Seq overview for all viral mapped reads (MR), as percent total reads (% Total). The number of unique BSJ variants and total BSJ counts is reported for high-confidence viral circRNAs. (B) Overlapping identity of high-confidence viral circRNAs, includes only BSJ with ≥10 reads per sample. (C, D) Visualization of high-confidence HSV-1 and KSHV circRNAs. Green and black traces include circular (back-spliced reads) and linear (non-chimeric) reads, respectively. Traces are the sum of raw BSJ or linear read values for biological duplicates. Y axis minimum and maximum values are shown on the right. Viral genes are shown below and labeled by gene class as IE (yellow), E (green), L1 (blue), and L2 (purple). Data Information: RNA-Seq (n = 2) was performed and high-confidence circRNAs were called using CHARLIE. In column bar graphs, data points are biological replicates, and bar maxima are the average.

