## [Peer Review File · The EMBO Journal]

Noncanonical circRNA biogenesis driven by alpha and gamma herpesviruses

Sarah Dremel, Vishal Koparde, Jesse Arbuckle, Chad Hogan, Thomas Kristie, Laurie T. Krug, Nicholas Conrad, and Joseph Ziegelbauer

Corresponding author(s): Joseph Ziegelbauer (ziegelbauerjm@mail.nih.gov)

Review Timeline:

Submission Date:	26th Jan 24
Editorial Decision:	5th Feb 24
Appeal Received:	22nd Mar 24
Editorial Decision:	19th Apr 24
Revision Received:	20th Dec 24
Editorial Decision:	28th Jan 25
Revision Received:	5th Feb 25
Accepted:	14th Feb 25

Editors: Kelly M Anderson and Cornelius Schneider

Transaction Report:

Dear Dr. Ziegelbauer,

Thank you for submitting your manuscript (EMBOJ-2024-116784) to The EMBO Journal. I have now had a chance to read it carefully and to discuss it with my colleagues, and I am sorry to say that we cannot offer publication in The EMBO Journal.

We appreciate that you have provided a resource of all the circRNAs in three species of herpesvirus during various phases of the viral life cycle. We believe this work will be a valuable resource to those in your field, particularly your finding that backsplicing of these circRNAs could be in part mediated by ORF57. However, considering circRNAs from other viruses have been shown to be backspliced independently of the spliceosome, we would require a more comprehensive understanding of how backsplicing is mediated, or some functional characterization of the circRNAs identified for further consideration in The EMBO Journal.

That said, we feel that your study may nevertheless be suitable for Life Science Alliance (<http://www.life-science-alliance.org/>), our open access journal launched in partnership with Rockefeller University Press and Cold Spring Harbor Laboratory Press and aiming to publish work of value to particular communities across all areas in biology. The editors of Life Science Alliance would be pleased to send your work for in-depth external review, without need for prior reformatting. Should you be interested in this option, please simply follow the link below for transfer.

I thank you for the opportunity to consider your manuscript and I am sorry that I can't be more positive on this occasion. I wish you every success in publishing this work and I hope you consider Life Science Alliance.

Kind regards,

Kelly M Anderson, PhD
Editor, The EMBO Journal
k.anderson@embojournal.org

**** As a service to authors, EMBO Press provides authors with the possibility to transfer a manuscript that one journal cannot offer to publish to another EMBO publication or the open access journal Life Science Alliance launched in partnership between EMBO Press, Rockefeller University Press and Cold Spring Harbor Laboratory Press. The full manuscript and if applicable, reviewers' reports, are automatically sent to the receiving journal to allow for fast handling and a prompt decision on your manuscript. For more details of this service, and to transfer your manuscript please click on Link Not Available. ****

Dear Dr. Anderson,

We thank you for your comments and analysis of our manuscript. You mentioned a requirement for "a more comprehensive understanding of how backsplicing is mediated" in your review. We recently obtained results that directly address your comment. We were planning on including these new results in a revised manuscript, but we can incorporate them now. In brief, the attached data shows that these viral circRNAs are not affected by depletion of the RNA lariat debranching enzyme (DBR1), which supports that these viral circRNAs are not RNA lariats. More importantly for your comment, we also show that expression of multiple viral circRNAs are decreased when either RLIG1 (RNA 5'-Phosphate And 3'-OH Ligase 1) or RTCB (RNA 2',3'-Cyclic Phosphate And 5'-OH Ligase) are repressed. This represents the first evidence for spliceosome-independent synthesis of these viral circRNAs for DNA viruses. This is also the first report of an RNA ligase-dependent transcript circularization in humans.

Given these new results that address your requirement, would you consider sending out this manuscript for review, if we incorporated this new information into our manuscript?

Respectively,
Joe

Dear Joe,

Thank you for submitting your manuscript for consideration by the EMBO Journal. As mentioned in our previous correspondence, the referees have been positive on the overall claims made in your manuscript but ask for significant additional work to more fully support those claims. Thank you also for providing a preliminary plan to address those concerns, which I feel is a reasonable plan to move forward.

Therefore, I would like to invite you to submit a revised version of the manuscript, addressing the comments of all three reviewers. I should add that it is EMBO Journal policy to allow only a single round of revision, and acceptance of your manuscript will therefore depend on the completeness of your responses in this revised version. That said, we are prepared to overrule the request to investigate the function of the circRNAs individually as we agree this can be left for a future manuscript.

Thank you for the opportunity to consider your work for publication. I look forward to your revision.

Yours sincerely,

Kelly M Anderson, PhD
Editor, The EMBO Journal
k.anderson@embojournal.org

Please remember: Digital image enhancement is acceptable practice, as long as it accurately represents the original data and conforms to community standards. If a figure has been subjected to significant electronic manipulation, this must be noted in the figure legend or in the 'Materials and Methods' section. The editors reserve the right to request original versions of figures and

the original images that were used to assemble the figure.

We realize that it is difficult to revise to a specific deadline. In the interest of protecting the conceptual advance provided by the work, we recommend a revision within 3 months (18th Jul 2024). Please discuss the revision progress ahead of this time with the editor if you require more time to complete the revisions. Use the link below to submit your revision:

Referee #1:

The Manuscript titled "Noncanonical circRNA biogenesis driven by alpha and gamma herpesviruses" by Dremel et al investigated the viral circRNAs from HSV-1-, KSHV- or MHV68-infected cells. They analyzed the viral circRNAs from different types of viral infections including lytic infection, latent infection and reactivation. The study is well-organized, and the figures were well presented. Overall, the presented data are interesting. This reviewer has some comments that need the authors to address.

Major questions:

1. The authors developed CHARLIE, which combines multiple circRNA identification tools. Has this tool been published? Has its performance and reliability been assessed? Different software algorithms may yield different results, leading to significant discrepancies. How does CHARLIE distinguish true positive back-splice junctions (BSJs) from false positives?
2. Can the experimentally validated Viral circRNA prove that CHARLIE has advantages over other circRNA identification tools that have already been published, such as CIRCexplorer, CIRI2, DCC, and circRNAFinder?
3. In Figure 1, it is intriguing that no viral circRNA was detected during the latent and reactivation stages of HSV-1 infection, and similarly, no viral circRNAs were identified during KSHV and MHV68 latency.

However, the authors should demonstrate the effectiveness of their latent infection and reactivation models. For instance, DNA FISH could be employed to visualize both RNA and DNA in latently infected cells and during reactivation.

Additionally, it is puzzling why the 122 HSV-1 circRNAs found at 12 hours post-infection (hpi) are not present at 24 hpi. Considering that circRNAs are generally considered stable, their disappearance raises questions about their fate and requires further investigation. This raises concerns about the accuracy of CHARLIE.

Similar concerns arise in Figure 4E, where the same virus infects the same cells, with the only difference being treatment with IgG at 1 hpi and 3 hpi, yet the circRNAs exhibit significant differences.

4. In Sup Fig. 1-4B, the authors only used divergent primer qPCR and RNase R for validation. It is necessary to confirm band specificity using RT-PCR followed by running an agarose gel and to confirm the BSJ interface of Viral circRNA using Sanger sequencing.
5. In circRNA-miRNA-mRNA analysis, the authors used Expected full-length sequences (sequence between 3' and 5' back-splice junctions) as the full-length sequences of circRNA, which may be incorrect. Not all mature circRNA full-length sequences are directly from 3' to 5', as there may be cases of intron-exon splicing [Ling-Ling Chen, Nat Cell Biol, PMID: 36658223, Zhang J, Nat Biotechnol, 2021, PMID: 33707777].
6. In Figure 2, the circRNAs are mapped onto the viral genome for three viruses during lytic infection. The authors should have provided a summary of the distribution of circRNA abundance across the corresponding viral genomes.
7. In Figure 3E, it is intriguing to investigate the interaction between RBPs and circRNAs. Could the authors conduct RNA-RBP precipitation assays to confirm the actual interaction between RBPs and circRNAs?
8. In Figure 4E, how do authors explain that 3 groups (Vehicle, IGG 1 hpi and IGG 3 hpi) of HSV-1 generate different viral circular RNAs? The same questions for those in PB for HSV-1 and IGG/PB and for KSHV.
9. In Figure 5, author found that RNA ligases may play roles in biogenesis of viral circRNAs. authors need to show the proteins of the RNA ligases (RLIG1 and RTCB) by the western blot assay. It would be the best if they can show the interaction between the ligases and viral circ RNAs.

Minor concerns:

1. Throughout the manuscript, RNA sequences are typed with "T" instead of "U", please correct them.
2. In Figure 1B, the Y-axis for KSHV and MHV68 is missing, and "human circRNA" is inaccurate for TG, which originates from mouse.

Referee #2:

The manuscript "Noncanonical circRNA biogenesis driven by alpha and gamma herpesviruses" by Dremel et al. present a systemic approach to identify and validate circRNAs during herpesvirus infection. The authors obtained RNA seq information from different herpesviruses (HSV-1, KSHV and MHV68) at different infection modes (lytic, latency and reactivation, in vitro and in vivo). They further characterized the viral circRNAs and identified potential host and viral proteins that might be involved in the formation of these viral circRNAs. This work has several interesting findings, especially as it is the first time (to the best of my knowledge) that circRNAs were identified in HSV-1. The manuscript is well written, and the results are presented clearly. While the results are novel and interesting, the lack of biological role for these new viral circRNAs, reduces the overall enthusiasm from this work.

Major issues:

1. Identifying the biological function of the identified viral circRNAs is required for confirming the results of the authors analysis. The authors have previously identified the role of circRELL1 (Dremel et al. EMBO Rep 2024) during HSV-1 infection. I think a similar approach is needed for at least some of the high incidence lytic circRNAs identified in this paper.
2. The authors suggest that RNA ligases depletion reduce the formation of several viral circRNAs. While this is a novel and interesting explanation for the formation of viral circRNAs, the presented experiments raise several issues.
 - a. To support this conclusion, a global approach for testing the circRNAs is needed (testing of only seven circRNAs allows only limited conclusion).
 - b. How these findings are supported by the double knockdown (supp figure 5-2) which seems to have even less effect compared to RTGB alone.
 - c. The effect of the RNA ligases depletion on viral gene expression and titers will also strengthen the conclusion (and the biological function of the circRNAs).

Minor points:

1. Line 58-60, only the association between HIV and HSV-2 was recognized. As HSV-2 is a different virus from HSV-1, the relevance to this paper is not clear.
2. Line 69-70 VZV has two approved vaccines.
3. The unique circPAN appearing in the ORF57 mutant (supp figure 7-3) and the elimination of all other circPAN with this mutant are puzzling. Has such a phenomenon ever appeared in any other sequence? Can the authors speculate of the role of such a mechanism?
4. Figure 3B the percent of the highly abundant D-A in the different viruses should be presented.
5. Figures 3C and 5C-H the Y axis should have the same scale to allow easier visualization of the differences.
6. uM and ug should be μM and μg throughout the paper.

Referee #3:

This manuscript describes a very detailed analysis of herpesvirus circRNA expression patterns which should represent a useful resource for researchers studying HSV-1, KSHV, or murine gammaherpesvirus 68. The authors have done extensive RNA-seq profiling and computational analyses, revealing that most viral circRNAs lack canonical splice donor-acceptor sites and are unlikely to be produced by the spliceosome. They instead show that some viral circRNAs require RNA ligases, which makes sense as it has previously been shown that circRNAs can be derived from tRNA introns using a similar mechanism. Future work is required to address the biological functions of the identified circRNAs but I think that is appropriately beyond the scope of this current manuscript. I only have minor comments and feel that this manuscript represents an important jumping off point that enables future efforts.

Minor points:

- The manuscript is overall very dense and the first part of the results section (P.15) in particular sounds more like a detailed methods section. The authors could work on adjusting the language and flow to make the main text more accessible.
- Line 127: The authors mention later in the text that the circRNAs are 100 fold less than GAPDH so it is too generous to say they "approach the abundance of the housekeeping gene GAPDH."

EMBOJ-2024-116784R-Q

Response to Reviewers

2024.09.12

Referee 1:

The Manuscript titled "Noncanonical circRNA biogenesis driven by alpha and gamma herpesviruses" by Dremel et al investigated the viral circRNAs from HSV-1-, KSHV- or MHV68-infected cells. They analyzed the viral circRNAs from different types of viral infections including lytic infection, latent infection and reactivation. The study is well-organized, and the figures were well presented. Overall, the presented data are interesting. This reviewer has some comments that need the authors to address.

Major questions:

1. The authors developed CHARLIE, which combines multiple circRNA identification tools. Has this tool been published? Has its performance and reliability been assessed? Different software algorithms may yield different results, leading to significant discrepancies. How does CHARLIE distinguish true positive back-splice junctions (BSJs) from false positives?

CHARLIE was developed for this manuscript and has not been previously published. A detailed description of CHARLIE has been included in the methods, and the github repository (<https://github.com/CCBR/CHARLIE>) is also publicly available. CHARLIE was developed for two reasons:

1. As the reviewer points out, different software algorithms may yield different results, which is why we wanted to look at BSJ calls which overlapped between the different published tools.
2. CHARLIE was developed to overcome concerns regarding accuracy and bias when applying published circRNA callers (CIRI2, CIRCexplorer2, DCC, circRNAfinder, find_circ) developed for large genomes of model organisms (e.g. mouse, human) to the smaller (viral) genome.

Performance and reliability of the individual circRNA calling tools has been published in Hansen et al. 2018 (PMID: 29556495). In this manuscript we report the incidence of CHARLIE high confidence circRNA calls which overlapped between biological replicates (Appendix Fig S2A) and overlapping circRNA calls between untreated and RNaseR digested sequencing material (Appendix Fig S2B). Additional information comparing BSJ calls and performance for the individual tools which CHARLIE aggregates is shown in the response below to reviewer #1 comment 2.

CHARLIE distinguishes true BSJs from false positives by the following:

In line with recent recommendations (PMID: 29556495, PMID: 32366901) for best circRNA practices we set the following requirements for *de novo* circRNA annotation. High confidence BSJ were identified by two distinct circRNA calling tools (CIRCexplorer2 + one other). As CIRI2 requires canonical splice donor-acceptor sequences—a bias we did not want when annotating viral circRNA—we included three additional tools (DCC, circRNAfinder, find_circ) lacking a splice D-A filter. High confidence BSJ calls also met the following criteria: ≥ 3 counts, 18 nucleotides (nt) mapped on both sides of the BSJ, and circRNA splice donor acceptor distance ≥ 150 nt. For paired end sequencing data, BSJ were further filtered, requiring the mate be contained within the 5' and 3' junction of the chimeric mapped read. To address concerns regarding higher error rates for RNA-Seq mapping to a small (viral) genome, we required high confidence circRNA be identified via CIRCexplorer2 in distinct mapping outputs (BWA and STAR). For viral circRNAs, splice donor-acceptor distance was limited to 5 kb to prevent artefactual calls caused by the isomeric structure of the HSV-1 genome.

2. Can the experimentally validated Viral circRNA prove that CHARLIE has advantages over other circRNA identification tools that have already been published, such as CIRCexplorer, CIRI2, DCC, and circRNAFinder?

Our opinion is that CHARLIE provides the following improvements to existing tools:

1. Compatible with a small (10-200 kb), poorly annotated (absence of TSS, TES, exons, introns) genome, such as viral genomes.
2. Does not require the presence of a canonical splice donor-acceptor to report a BSJ
3. Additional stringency filters, which result in filtering of false-positives

Response to Reviewers

Below is a comparison of calls made amongst these tools for untreated and RNase R-digested RNA-Sequencing from lymphatic endothelial cells (LEC) infected with KSHV for 3 days. After applying our initial filter which requires agreement in calls made by CIRCExplorer using STAR and BWA bam outputs, we observe significant overlap of BSJ calls from other tools. >90% KSHV circRNA and >95% human circRNA were called by ≥ 2 tools after requiring CIRCExplorer agreement.

Using CHARLIE we compared high confidence human circRNA identified in RNA-Seq data for our human lymphatic endothelial cells (LEC) infected with KSHV. 86% (RNase R-digested) and 82% (untreated) of high confidence human circRNA were annotated in either circAtlas or circBase.

Comparison of annotated BSJ species to those found using CHARLIE: LEC 72 hpi RNA-Seq

- circBase (<http://www.circbase.org/>)
n=768986 BSJ
- circAtlas 3.0 (<https://ngdc.cncb.ac.cn/circatlas/>)
n=92198 BSJ
- KSHV infected LEC
n=2664 BSJ
- KSHV infected LEC, RNase R treated
n=4334 BSJ

3. In Figure 1, it is intriguing that no viral circRNA was detected during the latent and reactivation stages of HSV-1 infection, and similarly, no viral circRNAs were identified during KSHV and MHV68 latency.

KSHV and MHV68 circRNAs were detected in latency models, see figure 1A (below). These viral circRNAs were also identified in lytic infection models, thus they are annotated in the middle of the venn diagrams. Echoing the limited transcriptional profile expressed during these modes of infection, we identified far fewer viral circRNAs as compared to lytic infection.

HSV-1 circRNAs were not detected in latency or reactivation models when using a requirement of 3 BSJ reads for a given circRNA. We relaxed these requirements and subsequently identified viral circRNA expressed from the viral latency gene (Fig EV1A-B). One of these variants [KT899744.1:6137-6814] was also identified in our lytic infection model. This circRNA was validated by divergent PCR followed by a gel check (Fig EV1C) or amplicon sequencing (Fig EV1D-E), as well as RNaseR digestion and divergent qPCR (Fig EV1F).

Response to Reviewers

However, the authors should demonstrate the effectiveness of their latent infection and reactivation models. For instance, DNA FISH could be employed to visualize both RNA and DNA in latently infected cells and during reactivation.

The models included in this manuscript follow expected patterns of behavior in regards to transcriptional profiles (Appendix Fig S1) and replication competence (additional details below).

HSV-1. The latency and reactivation models have been previously validated by the Kristie lab as published in Alfonso-Dunn *et al.* 2017 *Cell Host & Microbe* (PMID: 28407486). Their validation included analysis of per ganglia viral proteins (immunofluorescence) and transcripts (qPCR) and total yield of viral genomes and progeny. Our RNA-Seq data (Appendix Fig S1A) demonstrates expression from only the LAT gene during latency (infected TG, infected TG explant + ACV), whereas reactivation (infected TG explant) results in expression of lytic genes including ICP22, ICP27, ICP47, US9, UL35 and UL44.

KSHV. We have previously validated the infection models via transcriptomics and quantification of infectious viral progeny, as published in Tagawa *et al.* 2023 *PNAS* Fig. S2 (PMID: 36724259). Additional information regarding viral replication (genomes, encapsidated particles, infectious particles) is shown below. Our RNA-Seq data (Appendix Fig S1B) demonstrates expression predominantly from the latency locus (T0.7) during latency (infected VEC), whereas our lytic models (infected LEC, iSLK-BAC16 + NaB/Dox) results in expression of lytic genes spanning all kinetic classes.

MHV68. We have previously validated the NIH 3T3 infection models via genome-wide transcriptomics by microarray in Figs. 1-2 in Cheng *et al.* 2012 *J. Virology* (PMID: 22318145), and quantification of viral genomes and infectious viral progeny in Fig. 2C of Van Skike *et al.* 2018 *PLoS Pathogens* (PMID: 29390024). The absence of virus replication by the ORF50.stop virus measured by plaque assay and direct visualization of YFP+ CPE was characterized in Fig. 2 of Li *et al.* 2017 *Scientific Reports* (PMID: 28287622) and genome-wide transcriptome in microarray in Fig. 1 of Bland *et al.* 2024 *NPJ Vaccines* (PMID: 38914546). The HE-RIT B cell latency/reactivation model was deeply characterized (genome-wide transcripts by microarray, genome copy number by qPCR, infectious particles by plaque assay) in Figs. 1-2 of Santana *et al.* 2017 *Pathogens* (PMID: 28212352). MHV68 latency in WT GC B cells was characterized by enumeration of YFP+ GC B cells by flow cytometry and imaging of YFP+ GC B cell in splenic tissues by confocal microscopy in Fig. 4 of Hogan *et al.* 2024 *mBio* (PMID: 38170993). Our RNA-Seq data (Appendix Fig S1C) demonstrates expression predominantly from ORF73 encoding muLANA in latent models (GC B cells 16 dpi and HE-RIT without induction), whereas our lytic models (infected NIH3T12, HE-RIT + Dox/TPA) results in expression of lytic genes spanning all kinetic classes.

Response to Reviewers

Additionally, it is puzzling why the 122 HSV-1 circRNAs found at 12 hours post-infection (hpi) are not present at 24 hpi. Considering that circRNAs are generally considered stable, their disappearance raises questions about their fate and requires further investigation. This raises concerns about the accuracy of CHARLE. Similar concerns arise in Figure 4E, where the same virus infects the same cells, with the only difference being treatment with IgG at 1 hpi and 3 hpi, yet the circRNAs exhibit significant differences.

Detection of circRNA by RNA-Seq is not infallible, it has the potential to—as the reviewer points out—detect PCR artifacts as false-positives. The other downside is that “accuracy” and “reproducibility” of circRNA detection via RNA-Seq will be heavily influenced by the number of biological replicates, sequencing depth, and representation of a given gene (viral vs. host, RnaseR enriched or undigested) in the total population. circRNAs are a relatively low abundance transcriptional class. Thus, a possible reason not all circRNAs detected at 12 hpi overlap with those at 24 hpi is sampling chance and that in a given RNA-Seq sample a different set of low abundance circRNAs was detected. We present the following analysis to support this. We compared HSV-1 BSJ calls (infected MRC-5 12 and 24 hpi) after additional filters, including: variants found in ≥ 2 biological replicates, ≥ 10 counts, or ≥ 25 counts. We saw a modest increase in overlapping BSJ hits when filtering for those found in ≥ 2 replicates (from 28% to 30% overlap). We see a more significant increase when filtering BSJ hits by a raw counts threshold (40% overlap if ≥ 10 counts; 57% overlap if ≥ 25 counts). We feel this analysis suggests that the non-overlapping BSJ variants between 12 and 24 hpi are not because the circRNA species change with time, but rather comprise low abundance variants in a dataset.

Because, we report overlap based on exact BSJ positions, rather than relative to a given locus the Venn diagram makes the population seem to be quite distinct. Below we show the actual positions of BSJ (all high-confidence viral BSJ calls). The loci where back splicing occur are very similar between 12 and 24 hpi, even if the exact chimeric BSJ (nucleotide resolution) identified was variable.

Venn Diagrams: HSV-1 BSJ genomic positions

Response to Reviewers

We agree with the reviewer that using Venn diagrams to compare circRNA profiles using exact BSJ positions is more confusing than helpful. For this reason we have updated the venn diagrams in Fig. 4 (new Appendix Fig S7A) to only include variants with ≥ 10 raw BSJ reads per sample.

We have also included genomic traces showing all BSJ-containing and linear reads for each condition (Appendix Fig S7B-C)

Response to Reviewers

4. In Sup Fig. 1-4B, the authors only used divergent primer qPCR and RNase R for validation. It is necessary to confirm band specificity using RT-PCR followed by running a agarose gel and to confirm the BSJ interface of Viral circRNA using Sanger sequencing.

We have performed BSJ confirmation for viral circRNAs via divergent PCR followed by gel check and amplicon sequencing, see below (Appendix Fig S3B-E, Fig EV1C-F). Additional information on circRNA amplicons for all divergent viral primer sets used in this study is located in Dataset EV2.

5. In circRNA-miRNA-mRNA analysis, the authors used Expected full-length sequences (sequence between 3' and 5' back-splice junctions) as the full-length sequences of circRNA, which may be incorrect. Not all mature circRNA full-length sequences are directly from 3' to 5', as there may be cases of intron-exon splicing [Ling-Ling Chen, Nat Cell Biol, PMID: 36658223, Zhang J, Nat Biotechnol, 2021, PMID: 33707777].

We agree that an assumption was made regarding the full-length sequence of viral circRNAs, this is a limitation of using Illumina SRS to determine circRNA identity. We would assert that the genome architecture of herpesviruses (>90% single exon genes) makes this assumption less concerning than for the host (<2% single exon genes). However, we have now added a line of text to clarify this technical limitation to readers.

Lines 250-251: “This analysis has the caveat that we used the entire sequence between the 5' and 3' BSJ and this may not be true if additional splicing events are present.”

6. In Figure 2, the circRNAs are mapped onto the viral genome for three viruses during lytic infection. The authors should have provided a summary of the distribution of circRNA abundance across the corresponding viral genomes.

Fig. 2 plots circRNA distribution relative to the viral genome using both sashimi plots and traces of the BSJ containing reads. We are unclear what the reviewer believes is lacking, if the current plots are not sufficient could the reviewer please provide additional clarity on what they are asking for.

7. In Figure 3E, it is intriguing to investigate the interaction between RBPs and circRNAs. Could the authors conduct RNA-RBP precipitation assays to confirm the actual interaction between RBPs and circRNAs?

Our RNA-RBP findings in Fig. 3 are presented as *in silico* predictions. For one RBP, KSHV ORF57, we have included IP data (eCLIP) as well as validation of its subsequent role in circRNA synthesis. We agree that it is an interesting avenue of further research and hope to expand on it in future studies.

8. In Figure 4E, how do authors explain that 3 groups (Vehicle, IGG 1 hpi and IGG 3 hpi) of HSV-1 generate different viral circular RNAs? The same questions for those in PB for HSV-1 and IGG/PB and for KSHV.

Spliceosome inhibition has upstream effects on gene expression as key viral factors are spliced, including HSV-1 elongation factor (ICP22) and KSHV RNA binding protein (ORF57), see Fig. 4B-C. For this reason we always present circRNA abundance as a relative value which is normalized to the linear transcript reads at a given locus (Fig. 4F, Appendix Fig S7D-E). This is to account for trickle down effects of gene expression perturbation. Regarding the major differences in unique circRNA species found in the different samples, refer to our explanation in Reviewer #1 Comment #3. We agree that the Venn diagrams in (old) Fig. 4E are confusing within this experimental context and have updated them, as well as adding a supplemental figure (Appendix Fig S7B-C) showing linear vs. BSJ reads for all conditions.

9. In Figure 5, author found that RNA ligases may play roles in biogenesis of viral circRNAs. authors need to show the proteins of the RNA ligases (RLIG1 and RTCB) by the western blot assay. It would be the best if they can show the interaction between the ligases and viral circ RNAs.

We have now included western blots for RLIG1 and RTCB demonstrating effective siRNA depletion at the protein level in our various models, see Appendix Fig S9C-D.

Due to the rate at which the ligation reaction would be expected to occur, we feel this is not technically feasible as the RNA-protein interaction is likely to be transient. After a literature search, we failed to identify a publication which performed RNA immunoprecipitation of RTCB. However, RTCB have been used for *in vitro* splicing assays to generate circRNAs from linear transcripts (PMID: 30962542, reviewed in PMID: 33662562), supporting the feasibility of our model. We agree that further experiments should probe the exact role of RNA ligases in catalyzing viral back splicing and intend to expand on this mechanism in future work.

Minor concerns:

1. Throughout the manuscript, RNA sequences are typed with "T" instead of "U", please correct them.

We have fixed this throughout the manuscript.

2. In Figure 1B, the Y-axis for KSHV and MHV68 is missing, and "human circRNA" is inaccurate for TG, which originates from mouse.

Thank you for catching these labeling issues, we have fixed them in the new manuscript.

Referee 2:

The manuscript "Noncanonical circRNA biogenesis driven by alpha and gamma herpesviruses" by Dremel et al. present a systemic approach to identify and validate circRNAs during herpesvirus infection. The authors obtained RNA seq information from different herpesviruses (HSV-1, KSHV and MHV68) at different in infection modes (lytic, latency and reactivation, in vitro and in vivo). They further characterized the viral circRNAs and identified potential host and viral proteins that might be involved in the formation of these viral circRNAs. This work has several interesting findings, especially as it is the first time (to the best of my knowledge) that circRNAs were identified in HSV-1. The manuscript is well written, and the results are presented clearly. While the results are novel and interesting, the lack of biological role for these new viral circRNAs, reduces the overall enthusiasm from this work.

Major issues:

1. Identifying the biological function of the identified viral circRNAs is required for confirming the results of the authors analysis. The authors have previously identified the role of circRELL1 (Dremel et al. EMBO Rep 2024) during HSV-1 infection. I think a similar approach is needed for at least some of the high incidence lytic circRNAs identified in this paper.

This work is absolutely something we plan to do in our next manuscript, but we feel is beyond the scope of the current manuscript which is focused on *de novo* circRNA identification and characterization of biogenesis.

2. The authors suggest that RNA ligases depletion reduce the formation of several viral circRNAs. While this is a novel and interesting explanation for the formation of viral circRNAs, the presented experiments raise several issues.

a. To support this conclusion, a global approach for testing the circRNAs is needed (testing of only seven circRNAs allows only limited conclusion).

As requested, we have performed RNA-Seq on samples from our RNA ligase depletion experiment. This data is now included in Fig. 5 and Fig EV5.

Using RNA-Seq, we found that RLIG1 depletion resulted in a statistically significant (p-value=0.002) decrease in HSV-1 circRNA abundance, when performing a loci-matched test for siNTC-siRLIG1. RTCB depletion did not have the same widespread effects. In the context of our KSHV model, we found that RTCB or RLIG1 depletion resulted in a statistically significant (p-value=0.04 or 0.006, respectively) decrease in viral circRNA abundance. We conclude that RTCB dependent enhancement was limited to select loci, including HSV-1 circICP0, circLAT, KSHV circPAN, circK2, and circK12. Our current data does not support RNA ligases as the machinery for all viral circRNAs—but does appear to enhance circularization for a subset of loci.

b. How these findings are supported by the double knockdown (supp figure 5-2) which seems to have even less effect compared to RTCB alone.

When we quantified protein expression for RTCB and RLIG1 in our models, we observed that the level of RTCB depletion in our HSV-1 model (MRC-5) was less than in the single knockdown.

	% remaining	
	RTCB	RLIG1
MRC-5 Uninf.		
siNTC	100%	100%
siRTCB	44%	92%
siRLIG1	170%	40%
siRTCB/RLIG1	77%	25%

Data from Appendix Fig S9:

	% remaining	
	RTCB	RLIG1
iSLK-BAC16 Uninf.		
siNTC	100%	100%
siRTCB	53%	94%
siRLIG1	130%	14%
siRTCB/RLIG1	52%	17%

We expect this may be why the impact on HSV-1 circRNAs are not as significant as the single knockdown. Furthermore, RTCB and RLIG1 frequently have inverse effects of KSHV circRNAs in our KSHV model, thus the double knockdown was not particularly informative. For these reasons we have removed the double knockdown condition (old Sup. Fig 5-2) from the paper.

c. The effect of the RNA ligases depletion on viral gene expression and titers will also strengthen the conclusion (and the biological function of the circRNAs).

We have now included data demonstrating the impact of RNA ligase depletion on HSV-1 and KSHV in Fig. 5 B-D, including quantitation of viral transcripts, genomes, and yield. We found that RNA ligase depletion resulted in mild alterations to viral transcription, including increased forward splicing (Fig 5C). RNA ligase depletion resulted in a significant decrease in HSV-1 viral yield and KSHV genome replication (Fig. 5D). RLIG1 depletion resulted in increased KSHV particles and infectious progeny (Fig. 5D). In summary, host RNA ligases play a role during infection, however the exact contribution (anti vs. pro-viral) is dependent on virus and RNA ligase.

Minor points:

1. Line 58-60, only the association between HIV and HSV-2 was recognized. As HSV-2 is a different virus from HSV-1, the relevance to this paper is not clear.

We have removed this line from the introduction.

2. Line 69-70 VZV has two approved vaccines.

We have adjusted the text to read, "Of the nine human herpesviruses, only varicella zoster virus has FDA-approved vaccines."

3. The unique circPAN appearing in the ORF57 mutant (supp figure 7-3) and the elimination of all other circPAN with this mutant are puzzling. Has such a phenomenon ever appeared in any other sequence? Can the authors speculate of the role of such a mechanism?

This is the only condition we tested which resulted in such a distinct loci-dependent shift in back splicing. We have now included additional text in the discussion regarding when/why in the viral life cycle such a shift in splicing may be important.

Lines 604-611: ORF57 dependent regulation of PAN may allow for modal expression, such that select species are expressed in abortive vs. replicative lytic infection. This may suggest the ORF57-suppressed species plays a role during abortive lytic infection. Alternatively, ORF57-enhanced species may be antiviral and the virus must suppress their expression until viral infection has progressed through key lytic check points (early gene expression, DNA replication). Further study of these circPAN modalities is necessary to evaluate their contributions during infection contexts.

4. Figure 3B the percent of the highly abundant D-A in the different viruses should be presented.

We have updated Fig. 3B as indicated.

5. Figures 3C and 5C-H the Y axis should have the same scale to allow easier visualization of the differences.

We have adjusted the axes in 3C and 5F as indicated.

6. uM and ug should be μM and μg throughout the paper.

We have fixed this throughout the manuscript.

Referee 3:

This manuscript describes a very detailed analysis of herpesvirus circRNA expression patterns which should represent a useful resource for researchers studying HSV-1, KSHV, or murine gammaherpesvirus 68. The authors have done extensive RNA-seq profiling and computational analyses, revealing that most viral circRNAs lack canonical splice donor-acceptor sites and are unlikely to be produced by the spliceosome. They instead show that some viral circRNAs require RNA ligases, which makes sense as it has previously been shown that circRNAs can be derived from tRNA introns using a similar mechanism. Future work is required to address the biological functions of the identified circRNAs but I think that is appropriately beyond the scope of this current manuscript. I only have minor comments and feel that this manuscript represents an important jumping off point that enables future efforts.

Minor points:

- The manuscript is overall very dense and the first part of the results section (P.15) in particular sounds more like a detailed methods section. The authors could work on adjusting the language and flow to make the main text more accessible.

As requested, we have streamlined the text regarding our profiling approach on page 15.

- Line 127: The authors mention later in the text that the circRNAs are 100 fold less than GAPDH so it is too generous to say they "approach the abundance of the housekeeping gene GAPDH."

This line has been removed from the revised manuscript.

Dear Dr Ziegelbauer,

Thank you for submitting a revised version of your manuscript. Your study has now been seen by all original referees, who find that their previous concerns have been addressed and now recommend publication of the manuscript. There remain only a few mainly editorial points that have to be addressed before I can extend formal acceptance of the manuscript:

- Please rename the Conflict-of-Interest section into "Disclosure and Competing Interests Statement", in accordance with our updated Guide to Authors (<https://www.embopress.org/competing-interests>)
- As we are switching from a free-text author contribution statement towards a more formal statement based on Contributor Role Taxonomy (CRediT) terms, please remove the present Author Contribution section and instead specify each author's contribution(s) directly in the Author Information page of our submission system during upload of the final manuscript. See <https://casrai.org/credit/> for more information.
- Please adjust the in-text callouts for individual figures and figure panels: e.g. 2A, 5A appears to be missing
- Please doublecheck the links for figures/tables in ToC of the APPENDIX 1 FILE as some of the links are not for the corresponding figures (<https://www.embopress.org/page/journal/14602075/authorguide#structuredmethods>)
- Please make sure to provide all the requested Source data files listed in the uploaded and attached Source Data checklist file, which you had been sent by my colleague Hannah Sonntag. Please complete the Source Data checklist and upload it to our online system. Source data files need to be saved in a scheme one figure/folder and then uploaded as .zip files. E.g. all the Source data files for figure 1 need to be saved in a single folder and this needs to be zipped and then uploaded as "SD figure 1.zip" file.
- Please provide suggestions for a short 'blurb' text prefacing and summing up the conceptual aspect of the study in two sentences (max. 250 characters), followed by 3-5 one-sentence 'bullet points' with brief factual statements of key results of the paper; they will form the basis of an editor-written 'Synopsis' accompanying the online version of the article. Please also provide an altered synopsis image, making sure that the aspect ratio conforms to our website's format - it should be exactly 550 pixels wide and between 300-600 pixels high.
- Figure Legends (main + EV):
 1. Please note that the exact p values are not provided in the legends of figures 4B, F; 5B, E, F; EV4 C, E.
 2. Please note that the box plots need to be defined in terms of minima, maxima, centre, bounds of box and whiskers, and percentile in the legends of figures 3C, 4F, 6B, D; EV3 A
 3. Please note that information related to n is missing in the legends of figures 3C, 4B, 5B, 6B, D
 4. Please note that the measure of center for the error bars needs to be defined in the legends of figures 4D, 5D-F; EV4 C-L
- Please adjust the order of the manuscript sections: Title page with complete author information, Abstract, Keywords, Introduction, Results, Discussion, Methods, Data Availability Section, Acknowledgements, Disclosure and Competing Interests Statement, References, Main figure legends, Tables, Expanded Figure Legends.
- Please rename EV figure legends in ms to Figure EV1-EV5
- Please provide an up-to-date email address for co-author Sarah E Dremel, so that acknowledgement emails from our office can be delivered to them.

With best regards,

Cornelius Schneider

Cornelius Schneider, PhD
Editor | The EMBO Journal
c.schneider@embojournal.org

We realize that it is difficult to revise to a specific deadline. In the interest of protecting the conceptual advance provided by the work, we recommend a revision within 3 months (28th Apr 2025). Please discuss the revision progress ahead of this time with the editor if you require more time to complete the revisions. Use the link below to submit your revision:

Referee #1:

The authors addressed all my concerns, I don't have further questions.

Referee #2:

I think this is much better version of the manuscript.
The authors have revised the manuscript and addressed most of our main concerns

Referee #3:

The authors have adequately addressed my prior concerns.

All editorial and formatting issues were resolved by the authors.

Dear Dr. Ziegelbauer,

I am pleased to inform you that your manuscript has been accepted for publication in the EMBO Journal.

Yours sincerely,

Cornelius Schneider, PhD
Editor
The EMBO Journal
c.schneider@embojournal.org
